# SMOOTHIE: SMOOTHING DIFFUSION ON TOKEN EMBEDDINGS FOR TEXT GENERATION

## ABSTRACT

Diffusion models have achieved state-of-the-art performance in generating images, audio, and video, but their adaptation to text remains challenging due to its discrete nature. Prior approaches either apply Gaussian diffusion in continuous latent spaces, which inherits semantic structure but struggles with token decoding, or operate in categorical simplex space, which respect discreteness but disregard semantic relation between tokens. In this paper, we propose Smoothing Diffusion on Token Embeddings (SMOOTHIE), a novel diffusion method that combines the strengths of both approaches by progressively smoothing token embeddings based on semantic similarity. This technique enables gradual information removal while maintaining a natural decoding process. Experimental results on several sequence-to-sequence generation tasks demonstrate that SMOOTHIE outperforms existing diffusion-based models in generation quality. Furthermore, ablation studies show that our proposed diffusion space yields better performance than both the standard embedding space and the categorical simplex.

## 1 INTRODUCTION

Diffusion models attracted a lot of attention in recent years as they show very high generation quality in image (Rombach et al., 2022; Podell et al., 2023), audio (Evans et al., 2024) and video (Blattmann et al., 2023) domains surpassing all previous approaches such as GANs (Goodfellow et al., 2014) and Normalizing Flows (Rezende & Mohamed, 2015). Diffusion models work by introducing a forward process that gradually degrades an object by injecting Gaussian noise into it, and then learning the reverse process by denoising the object.

Applying diffusion models to text is challenging due to its discrete nature. Nevertheless, several works have explored ways to design suitable diffusion processes. One line of research proposes gradually removing information by replacing tokens with others sampled from a categorical distribution (Austin et al., 2021; He et al., 2023; Lou et al., 2024). Another approach applies Gaussian diffusion to the latent space of token embeddings (Li et al., 2022; Gong et al., 2023a). Additionally, some studies leverage the discreteness of text by performing diffusion directly on the vocabulary probability simplex instead of the embedding space (Karimi Mahabadi et al., 2024; Han et al., 2023).

Each of the described methods offers distinct advantages and limitations, as summarized in Table 1. Gaussian diffusion progressively removes semantic information: under the Euclidean semantic space hypothesis (Hashimoto et al., 2016), the distinguishability of noisy tokens depends on their initial distances in the latent space. The addition of Gaussian noise gradually disrupts these distances, making the semantics of a latent representation increasingly difficult to recover. However, Gaussian diffusion does not account for the discrete nature of text, which complicates the mapping of generated latent vectors back to discrete tokens (Li et al., 2022; Shabalin et al., 2025).

On the other hand, categorical and simplex-based diffusion methods naturally preserve the discreteness of text and eliminate the need for an explicit decoding step. Nevertheless, they disregard semantic relationships between tokens during the noising process, resulting in a more erratic and less meaningful degradation of information.

In this paper, we propose SMOOTHIE, a smoothing diffusion framework that satisfies both properties. We represent each token with a vector based on distances between token embeddings. During the forward process, our diffusion mechanism gradually perturbs these distances, progressively dissolving

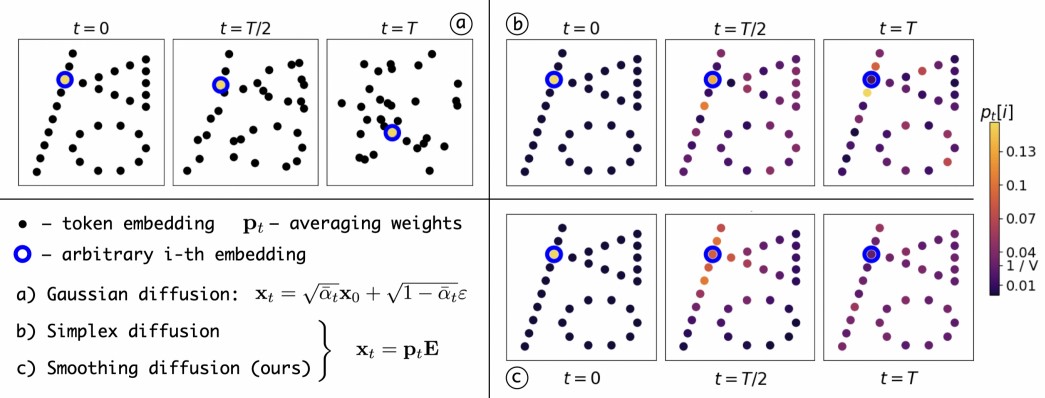

Figure 1: An illustration of the diffusion process for Gaussian, simplex, and smoothing diffusion methods. The key distinction between simplex and smoothing diffusion is that the latter incorporates semantic relationships between tokens during the noise addition process.

Table 1: Comparison of diffusion methods in terms of accounting for text discreteness and semantics.

|  | Categorical | Gaussian | Simplex | Smoothing (Ours) |
|---|:---:|:---:|:---:|:---:|
| Accounting for Discreteness | ✓ | ✗ | ✓ | ✓ |
| Accounting for Semantics | ✗ | ✓ | ✗ | ✓ |

semantic information. Like simplex diffusion, our method enables natural decoding from latent representations back to tokens. In theory, SMOOTHIE is applicable not only to text, but to any domain where data comes from a categorical distribution with inherent similarity between categories (e.g. graphs).

We evaluate SMOOTHIE on one unconditional and four sequence-to-sequence generation tasks and show that it outperforms existing diffusion-based approaches. Ablation studies further demonstrate that our method enables effective control over the trade-off between fluency and diversity of the generated text.

The main contributions of our work are as follows:

1. We propose a novel text diffusion framework that simultaneously respects the discrete nature of text and progressively removes semantic information from token representations during the forward process.

2. We show the practical effectiveness of our approach across multiple text generation tasks, providing empirical evidence for the advantages of our diffusion design.

## 2 PROBLEM STATEMENT AND BACKGROUND

**Problem statement** In this work, we develop a model for both unconditional and sequence-to-sequence generation tasks. In all cases, the objective is to generate a target sequence $\mathbf{w}^y = w_1^y, \ldots, w_m^y$. For sequence-to-sequence generation, the model additionally conditions on a source sequence $\mathbf{w}^x = w_1^x, \ldots, w_n^x$. We assume access to parallel datasets, where each source sequence is paired with its corresponding target sequence.

**Gaussian diffusion model** The diffusion process is defined in terms of a forward (noising) and a reverse (denoising) processes. Given an initial data point sampled from the data distribution, $\mathbf{x}_0 \sim p_{\text{data}}$, the forward process generates a sequence of progressively noisier latent variables $\mathbf{x}_1, \ldots, \mathbf{x}_T$. Each step in this sequence is defined by the transition $\mathbf{x}_t \sim q(\mathbf{x}_t \mid \mathbf{x}_{t-1}) = \mathcal{N}(\sqrt{\alpha_t}\mathbf{x}_{t-1}, \sqrt{1 - \alpha_t}, \varepsilon)$, where the parameter $\alpha_t \in [0, 1)$ controls the amount of noise injected at timestep $t$. This formu-

lation also supports a direct sampling of $\mathbf{x}_t$ from $\mathbf{x}_0$ using the marginal distribution $q(\mathbf{x}_t \mid \mathbf{x}_0) = \mathcal{N}(\sqrt{\bar{\alpha}_t}\mathbf{x}_0, \sqrt{1 - \bar{\alpha}_t}, \varepsilon)$, where $\bar{\alpha}_t = \prod_{s=0}^{t} \alpha_s$ denotes the cumulative product of noise scales.

After the forward process is complete, a neural network $f_\theta$ is trained to reverse it by predicting the original data point $\mathbf{x}_0$ from the noisy input $\mathbf{x}_t$. During generation, the model iteratively denoises an initial sample $\mathbf{x}_T \sim \mathcal{N}(0, I)$, gradually reconstructing the data through the learned reverse process until it recovers $\mathbf{x}_0$.

**Embedding diffusion**    The most popular continuous text diffusion approaches create a latent space by mapping tokens to their embeddings (Li et al., 2022; Gong et al., 2023a; Yuan et al., 2022). Then the Gaussian diffusion process is used to corrupt a latent. The decoding is usually performed by mapping a generated embedding to the token corresponding to the closest embedding.

**Simplex diffusion**    SSD-LM (Han et al., 2023) and TESS (Karimi Mahabadi et al., 2024) propose a simplex diffusion model. They map each token $w$ to a $k$-logit simplex $\mathbf{s}^w \in \{\pm k\}^V$, where $V$ is the size of the vocabulary and

$$\mathbf{s}_{(i)}^w = \begin{cases} +k, & i = w \\ -k, & \text{otherwise} \end{cases} \tag{1}$$

Then the latent is represented as a sequence $\mathbf{S}_0 = (\mathbf{s}^{w_1^y}, \ldots, \mathbf{s}^{w_m^y})$. Corruption is performed with the Gaussian diffusion process with noise variance multiplied by $k^2$ ($k = 5$ by default), $\mathbf{S}_t = \sqrt{\bar{\alpha}_t}\mathbf{S}_0 + k\sqrt{1 - \bar{\alpha}_t}\varepsilon$. The model input is calculated by first producing a probability simplex over vocabulary, $\mathbf{p}_t = \mathrm{softmax}(\mathbf{S}_t)$, and then averaging token embeddings with obtained weights, $\mathbf{p}_t\mathbf{E}$, where $\mathbf{E}$ is a matrix of token embeddings.

## 3 RELATED WORK

Since the initial attempt to apply diffusion models to text generation (Hoogeboom et al., 2021), numerous studies have explored ways to better align the diffusion process with the specifics of textual data. D3PM (Austin et al., 2021) tried exploiting the semantic property of tokens by applying a discrete diffusion process that replaces tokens with semantically similar alternatives with higher probability. However, their experiments showed that simple token masking approach produces better empirical results.

Diffusion-LM (Li et al., 2022) proposed applying Gaussian diffusion in the continuous latent space of token embeddings, while TEncDM (Shabalin et al., 2025) further demonstrated that context-dependent embeddings provide a more suitable latent space for continuous diffusion. Despite achieving strong generation quality, the downside of these methods is the requirement of an additional latent decoding step.

DiffuSeq-v2 (Gong et al., 2023b) attempted to bridge the gap between discrete and continuous diffusion models by combining masking with Gaussian noise during the noising process. Another research direction (Han et al., 2023; Karimi Mahabadi et al., 2024) focuses on mapping tokens to almost-one-hot simplex representations over the vocabulary and introducing Gaussian noise directly into this space. While this approach does not account for token semantics during noising, it preserves the discrete structure of text.

Our work is inspired by a different line of research developed in the image domain (Rissanen et al., 2023; Hoogeboom & Salimans, 2023), where semantic information is gradually removed by smoothing pixel values according to the heat dissipation principle. However, while being effective for continuous signals such as images, this strategy can not be directly applied to text due to its inherently discrete nature.

## 4 SMOOTHING DIFFUSION

In this section, we introduce SMOOTHIE, a smoothing text diffusion model that incorporates both the discrete nature of text and the semantic relationships between tokens into the diffusion process. We will first derive the diffusion process for unconditional generation and then extend it to conditional

generation. We provide an intuitive illustration of our approach, along with pseudo-code for the training and sampling procedures, in Fig. 1, Alg. 1, and Alg. 2, respectively.

## 4.1 FORWARD DIFFUSION PROCESS

Let $V$ denote the vocabulary size, and let $\mathbf{E} \in \mathbb{R}^{V \times d}$ be a fixed embedding matrix, where each row corresponds to a $d$-dimensional token embedding. To construct a latent space suitable for diffusion, we represent each token $w_i^y$ in a target sequence $\mathbf{w}^y$ with a vector of negative squared Euclidean distances between an embedding of token $w_i^y$ and embeddings of all tokens in the vocabulary:

$$\mathbf{D}_0 = \mathbf{D}_0(\mathbf{E}_{\mathbf{w}^y}) = \left\{ -\frac{\|\mathbf{E}_{w_i^y} - \mathbf{E}_j\|^2}{2} \right\}_{i,j=1}^{m,V} \tag{2}$$

Here, $\mathbf{E}_{w_i^y}$ is the embedding of the i-th token in the sequence, and $\mathbf{E}_j$ is the embedding of the j-th vocabulary token. To generate a trajectory of progressively noisier latents, we define a non-Markovian forward, or noising process:

$$\text{Forward process} \qquad q(\mathbf{D}_{1:T}|\mathbf{D}_0) = \prod_{t=1}^{T} q(\mathbf{D}_t|\mathbf{D}_0) = \prod_{t=1}^{T} \mathcal{N}\left(\mathbf{D}_t \middle| \frac{1}{\sigma_t^2}\mathbf{D}_0, \delta^2 I\right) \tag{3}$$

The noise scheduler $\sigma_t$ ($1 < \sigma_1 < \cdots < \sigma_T$) controls the amount of noise added at each timestep. The hyperparameter $\delta$ controls the stochasticity of the diffusion process and makes it non-deterministic. Following Rissanen et al. (2023), we keep $\delta$ independent of the timestep $t$.

To construct the model input, we convert $\mathbf{D}_t$ into a probability distribution over the vocabulary using the softmax function: $\mathbf{p}_t = \text{softmax}(\mathbf{D}_t)$. In this formulation, each token is represented by the weights of Nadaraya-Watson kernel estimator applied over all embeddings in the vocabulary with Gaussian kernel whose bandwidth is defined by $\sigma_t$. The choice of a Gaussian kernel is motivated by the Euclidean semantic space hypothesis (Hashimoto et al., 2016), which assumes that semantic similarity correlates with Euclidean proximity in embedding space. As a result, as $\sigma_t$ increases, the probability mass—initially centered in a single token—gradually distributes between all other tokens, starting from the most semantically similar and ending with the most distant ones (see Fig. 1 (c)).

Note that our approach can be viewed as a generalization of a simplex-based diffusion (Han et al., 2023; Karimi Mahabadi et al., 2024). In particular, by replacing our Euclidean distance with trivial metric, we get the latent space formulation defined in Eq. 1, which ignores the semantic relationships between tokens. We prove this statement in Appendix C. In Section 5 we show that incorporating semantic similarity into the diffusion process is crucial for achieving better performance.

## 4.2 REVERSE DIFFUSION PROCESS

The reverse, or denoising process, starts with a sample from prior distribution $p(\mathbf{D}_T)$ and ends with the denoised data sample $\mathbf{D}_0$. We define it as a Markov chain with Gaussian distributions:

$$\text{Reverse process} \quad p_\theta(\mathbf{D}_{0:T}) = p(\mathbf{D}_T) \prod_{t=1}^{T} p_\theta(\mathbf{D}_{t-1}|\mathbf{D}_t) = p(\mathbf{D}_T) \prod_{t=1}^{T} \mathcal{N}\big(\mathbf{D}_{t-1}|\mu_\theta(\mathbf{p}_t, t), \tilde{\delta}^2 I\big), \tag{4}$$

where $\theta$ are trainable model parameters and $\tilde{\delta}^2$ is a noise variance used in the reverse process. Inspired by Rissanen et al. (2023), we allow noise variance to change between the forward and reverse processes. That permits us to explicitly control the stochasticity of the generation trajectory, which significantly affects the model performance (see Section 5.1).

Our goal is to find such parameters $\theta$, that minimize the marginal negative likelihood of data samples $p_\theta(\mathbf{D}_0) = \int p_\theta(\mathbf{D}_{0:T}) \mathrm{d}\mathbf{D}_{1:T}$. We optimize the negative log-likelihood by minimizing its variational

upper bound:

$$-\log p_\theta(\mathbf{D}_0) = -\log \int \frac{p_\theta(\mathbf{D}_{0:T})q(\mathbf{D}_{1:T}|\mathbf{D}_0)}{q(\mathbf{D}_{1:T}|\mathbf{D}_0)}d\mathbf{D}_{1:T} \leq -\mathbb{E}_q \log \frac{p_\theta(\mathbf{D}_{0:T})}{q(\mathbf{D}_{1:T}|\mathbf{D}_0)} \tag{5}$$

$$= -\mathbb{E}_q \left[ \log \frac{p_\theta(\mathbf{D}_T)}{q(\mathbf{D}_T|\mathbf{D}_0)} + \sum_{t=2}^{T} \log \frac{p_\theta(\mathbf{D}_{t-1}|\mathbf{D}_t)}{q(\mathbf{D}_{t-1}|\mathbf{D}_0)} + \log p_\theta(\mathbf{D}_0|\mathbf{D}_1) \right] \tag{6}$$

$$= \mathbb{E}_q \left[ \underbrace{\mathrm{D}_{\mathrm{KL}}\big[q(\mathbf{D}_T|\mathbf{D}_0)\|p(\mathbf{D}_T)\big]}_{L_T} + \sum_{t=2}^{T} \underbrace{\mathrm{D}_{\mathrm{KL}}\big[q(\mathbf{D}_{t-1}|\mathbf{D}_0)\|p_\theta(\mathbf{D}_{t-1}|\mathbf{D}_t)\big]}_{L_{t-1}} \underbrace{-\log p_\theta(\mathbf{D}_0|\mathbf{D}_1)}_{L_0} \right]$$
$$\tag{7}$$

In this formula, $L_T$ is constant during the training, as it does not depend on any learnable parameters. Both forward and reverse processes are defined by Gaussian distributions, which allows us to compute the terms $L_0$ and $L_{t-1}$ in closed form:

$$L_0 = \mathbb{E}_q \left[ \frac{1}{2\tilde{\delta}^2} \|\mathbf{D}_0 - \mu_\theta(\mathbf{p}_1, 1)\|^2 \right] + C_0; \; L_{t-1} = \mathbb{E}_q \left[ \frac{1}{2\tilde{\delta}^2} \left\| \frac{1}{\sigma_t^2}\mathbf{D}_0 - \mu_\theta(\mathbf{p}_t, t) \right\|^2 \right] + C_{t-1}, \tag{8}$$

where $C_0$ and $C_{t-1}$ are constants that do not depend on parameters $\theta$. This implies that the most direct parameterization of $\mu_\theta$ is a model that predicts $\mathbf{D}_0/\sigma_t^2$, corresponding to the posterior mean of the forward process. However, for practical reasons, we instead parameterize $\mu_\theta$ as $g_\theta/\sigma_t^2$ which ensures that all model outputs are scaled to have the same variance across timesteps.

$$L_{t-1} = \mathbb{E}_q \left[ \frac{1}{2\tilde{\delta}^2\sigma_t^4} \|\mathbf{D}_0 - g_\theta(\mathbf{p}_t, t)\|^2 \right] + C_{t-1}, \tag{9}$$

Following Ho et al. (2020), we replace $L_{t-1}$ with its simplified version by removing the scaling coefficient $2\tilde{\delta}^2\sigma_t^4$, resulting in the following loss function:

$$L_\mathbf{D}(\theta) = \mathbb{E}_{\mathbf{w}^y, t, \mathbf{p}_t} \left[ \|\mathbf{D}_0(\mathbf{E}_{\mathbf{w}^y}) - g_\theta(\mathbf{p}_t, t)\|^2 \right] \tag{10}$$

However, this loss function is challenging to optimize due to the high variance and dimensionality of $\mathbf{D}_0$. To address this issue, we introduce the following theorem:

**Theorem 4.1.** *Let $g^*(\mathbf{p}_t, t)$ be an optimal prediction for Eq. 10. Then $g^*(\mathbf{p}_t, t) = \mathbf{D}_0(f^*(\mathbf{p}_t, t)) + C$, where $C$ is a constant that does not depend on $f^*(\mathbf{p}_t, t)$ and $f^*(\mathbf{p}_t, t)$ is an optimal prediction for Eq. 11*

$$L_\mathbf{E}(\theta) = \mathbb{E}_{\mathbf{w}^y, t, \mathbf{p}_t} \left[ \|\mathbf{E}_{\mathbf{w}^y} - f_\theta(\mathbf{p}_t, t)\|^2 \right] \tag{11}$$

We train the model $f_\theta$ by minimizing Eq. 11. During the sampling, we initialize from $\mathbf{D}_T \sim \mathcal{N}(0, \tilde{\delta}^2 I)$ and iteratively update it over 200 steps using the following scheme:

$$\mathbf{D}_{t-1} = \frac{1}{\sigma_{t-1}^2}\mathbf{D}_0(f_\theta(\mathbf{p}_t, t)) + \tilde{\delta}\varepsilon, \tag{12}$$

Note that by Th. 4.1, this procedure is equivalent to updating $\mathbf{D}_{t-1}$ as $\mathbf{D}_{t-1} = g_\theta(\mathbf{p}_t, t)/\sigma_{t-1}^2 + \tilde{\delta}\varepsilon$, where $g_\theta$ is optimized with Eq. 10, because models take $\mathbf{p}_t = \mathrm{softmax}(\mathbf{D}_t)$ as input, which is invariant to shifts of $\mathbf{D}_t$. The proof of Th. 4.1 is provided in Appendix D.

In contrast, related methods such as SSD-LM (Han et al., 2023) and TESS (Karimi Mahabadi et al., 2024) employ cross-entropy loss during training. While our method is also compatible with this loss, in our experiments it led to inferior performance and faster overfitting. Therefore, we chose to rely on the MSE objective.

---

**Algorithm 1** Training

> **Input:** $\mathbf{w}^x, \mathbf{w}^y, \delta, t \sim \mathcal{U}(1,T), \varepsilon \sim \mathcal{N}(0,I)$
> Compute $\mathbf{D}_0$ with Eq. 2
> Compute $\mathbf{D}_t = \mathbf{D}_t/\sigma_t^2 + \delta\varepsilon$
> Compute $\mathbf{p}_t = \text{softmax}(\mathbf{D}_t)$
> Minimize $\|\mathbf{E}_{\mathbf{w}^y} - f_\theta(\mathbf{p}_t, t, \mathbf{w}^x)\|^2$

---

**Algorithm 2** Sampling

> **Input:** Source text $\mathbf{w}^x$, model $f_\theta$, noise std $\tilde{\delta}$
> Sample $\mathbf{D}_T \sim \mathcal{N}(0, \tilde{\delta}^2 I)$
> **for** $t$ in $\{T, \ldots, 1\}$ **do**
>     Compute $\mathbf{p}_t = \text{softmax}(\mathbf{D}_t)$
>     Compute $\mathbf{D}_{t-1}$ with Eq. 12
> **end for**
> Decode tokens $\hat{\mathbf{w}}^y = \text{argmax}(\mathbf{D}_0)$

---

### 4.3 NOISE SCHEDULER

The noise scheduler plays a crucial role in the diffusion process by controlling the rate at which the signal decays over time. Following the observation that text diffusion models benefit from adding more noise at the early stages of the forward process (Shabalin et al., 2025), we define our noise schedule as follows:

$$\sigma_t = (\sigma_{\max} - \sigma_{\min})\frac{2}{\pi}\arctan\left(\frac{1}{d}\sqrt{\frac{t}{T-t+\epsilon}}\right) + \sigma_{\min}, \quad \forall t \in [0,T] \tag{13}$$

Here, $\sigma_{\min}$ and $\sigma_{\max}$ sets the minumum and maximum bandwidth respectively, $d$ controls the rate of noise accumulation, and $\epsilon$ is a small constant added to prevent division by zero. Throughout our experiments, we use $\sigma_{\min} = 1.5$, $\sigma_{\max} = 200$ and $d \in \{5,7\}$ to achieve a linear increase in model entropy with increasing $t$ (Dieleman et al., 2022). Also, we set $\delta = 1$ during training. We discuss the noise scheduler ablation in Appendix I.

### 4.4 SELF-CONDITIONING

Following previous works (Dieleman et al., 2022; Shabalin et al., 2025; Karimi Mahabadi et al., 2024), we employ *self-conditioning* (Chen et al., 2023) to our model. During training, with 50% probability the model is fed with self-condition set to zero: $\hat{\mathbf{x}}_0^t = f_\theta(\mathbf{p}_t, \mathbf{0}, t)$. Otherwise the model receives its previous prediction as an input: $\hat{\mathbf{x}}_0^t = f_\theta(\mathbf{p}_t, \text{SG}(\bar{\mathbf{x}}_0^t), t)$, where $\bar{\mathbf{x}}_0^t = f_\theta(\mathbf{p}_t, \mathbf{0}, t)$ and SG is the stop-gradient function that prevent gradients from flowing through $\bar{\mathbf{x}}_0^t$. During the generation stage, the first prediction is made with self-condition set to zero and at all subsequent steps the predictions are performed as $\hat{\mathbf{x}}_0^t = f_\theta(\mathbf{p}_t, \hat{\mathbf{x}}_0^{t+1}, t)$. We demonstrate the impact of self-conditioning in Appendix G.

### 4.5 SEQUENCE LENGTH

Because diffusion models operate over fixed-length sequences, we pad all shorter sequences using a special padding token, which the model is trained to predict. To limit computational overhead, we set the maximum sequence length for each dataset to approximately the 99th percentile of training set sequence lengths. The exact values used for each dataset are provided in the Appendix E.

## 5 EXPERIMENTS

**Implementation details** In all experiments, we use a pre-trained embedding matrix, $\mathbf{E}$, from the BERT (Devlin et al., 2019) model[1]. We normalize this matrix to have a zero mean and a unit variance and keep it fixed throughout training. Although the model receives the soft token distribution $\mathbf{p}_t$ as input, it does not operate directly on this distribution. Instead, we compute a weighted average of the token embeddings, $\mathbf{p}_t\mathbf{E}$, which yields a lower-dimensional, more tractable representation for the model to process.

Our model architecture is based on the design proposed in Shabalin et al. (2025), consisting of Transformer decoder layers (Vaswani et al., 2017) augmented with UNet-style skip connections.

---

[1]We discuss the ablation of other embedding types in Appendix H

Specifically, the output of the first layer is added to the input of the last, the second to the second-last, and so on. The full model has 12 layers and approximately 100M parameters. For conditional generation, we modify the model to accept an input sequence $\mathbf{w}^x$, which is processed by an additional 6-layer Transformer encoder. The encoder output is integrated into the decoder through cross-attention mechanisms. For timestep conditioning, we adopt the approach from Gong et al. (2023a), plugging learned timestep embeddings into each Transformer block akin to positional embeddings. The complete set of hyperparameters used for training and evaluation is provided in Appendix E.

## 5.1 THE IMPORTANCE OF $\tilde{\delta}$

Before presenting results on seq-to-seq generation tasks, we highlight the importance of the hyperparameter $\tilde{\delta}$, which controls the stochasticity of the denoising process. To illustrate its impact, we evaluate generation quality on an *unconditional* generation task using different values of $\tilde{\delta}$. Specifically, we use the **ROCStories** dataset and assess performance using three metrics: generative **perplexity** (to estimate average text quality), **diversity** (to measure lexical variety) (Su et al., 2022), and the **MAUVE Score** (Pillutla et al., 2021) (to evaluate the overall similarity of generated texts to the reference distribution). When calculating MAUVE, we generate 1,000 texts five times with different seeds and compare them with 1,000 randomly sampled reference texts. We then average the results.

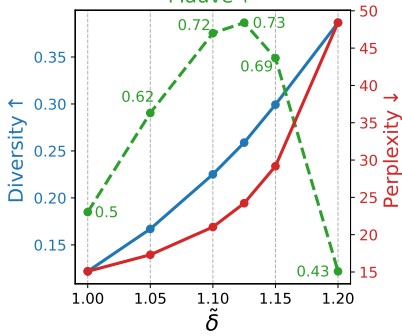

Figure 2: Unconditional generation quality for $\delta = 1$ and varying $\tilde{\delta}$.

Figure 2 shows the results for a model trained with $\delta = 1$. We observe that lower values of $\tilde{\delta}$ lead to better perplexity scores but lower diversity. In other words, reduced stochasticity improves the quality of individual texts but decreases their uniqueness. This trade-off is actually desirable for sequence-to-sequence tasks, where diversity typically arises naturally from the varying input conditions. In Appendix F, we justify this insight by grid-searching the best $\tilde{\delta}$ value. As a result, we set $\tilde{\delta} = 0.1$ for all sequence-to-sequence experiments.

In contrast, for unconditional generation, the optimal value of $\tilde{\delta}$ is slightly higher than the one used during training, as indicated by the MAUVE Score. At this point, the generated texts exhibit sufficient diversity while maintaining acceptable perplexity. These findings show that $\tilde{\delta}$ has a strong influence on the generation process and should be tuned carefully depending on the target task.

**Datasets** In addition to the unconditional generation on **ROCStories** dataset, we evaluate SMOOTHIE on four sequence-to-sequence datasets of varying difficulty. For *paraphrase generation*, we use the Quora Question Pairs **(QQP)** dataset (Chen et al., 2017), which contains 147K pairs of semantically equivalent questions. For *question generation*, we adopt the **Quasar-T** dataset (Dhingra et al., 2017), processed by Gong et al. (2023a), resulting in 119K document-question pairs. For *summarization*, we use the **XSum** dataset (Narayan et al., 2018), comprising 204K BBC articles and their corresponding summaries. For *detoxification*, we use **ParaDetox** (Logacheva et al., 2022) dataset with 19,766 pairs of toxic and neutral comments. More detailed information about each dataset is provided in the Appendix L.

**Metrics** Following the evaluation protocol from Gong et al. (2023a); Karimi Mahabadi et al. (2024), we employ a combination of n-gram-based, diversity and semantic similarity metrics. Specifically, we report **BLEU** (Papineni et al., 2002) and **ROUGE-1/2/L** (Lin, 2004) scores to measure lexical overlap between generated and reference texts, and **BERTScore (BS)** (Zhang et al., 2020) to assess semantic similarity. For BERTScore, we use the `microsoft/deberta-xlarge-mnli` model to ensure consistency with previous studies (Yuan et al., 2022; Karimi Mahabadi et al., 2024).

To evaluate the diversity of generated texts, we compute n-gram diversity (Deshpande et al., 2019), which reports the fraction of unique unigrams (**Div-1**) and 4-grams (**Div-4**) in a text. Additionally, for the text detoxification task, we measure **J-Score**, which comprises text fluency, style accuracy, and content preservation.

Table 2: Results on XSum (left) and Quasar-T (right) datasets. † denotes autoregressive models, △ denotes the results reproduced with original code, ⋆ denotes our implementations. The best-performing *diffusion* results are highlighted in **bold**, the second-best are underlined.

| | XSum | | | Quasar-T | | | |
|---|---|---|---|---|---|---|---|
| Method | BS ↑ | R-1/2/L ↑ | Method | BS ↑ | BLEU ↑ | R-L ↑ | D-1/4 |
| GPT-2†△ | 69.0 | 28.3/8.2/21.8 | GPT-2† | 60.5 | 7.4 | 27.2 | 96.0/92.2 |
| Transformer† | — | 30.5/10.4/24.2 | GPVAE-T5† | 63.1 | 12.5 | 33.9 | 93.8/72.8 |
| FLAN-T5† | 72.7 | 34.6/12.9/27.2 | BART† | 66.2 | 17.4 | 38.8 | 98.2/61.7 |
| MDLM△ | 62.1 | 27.9/7.7/21.1 | MDLM△ | 60.7 | 17.5 | 33.6 | 91.0/**64.2** |
| DiffuSeq | 46.8 | 18.9/1.3/13.6 | DiffuSeq | 59.4 | 15.8 | — | 91.1/— |
| SeqDiffuSeq△ | 61.8 | 28.6/6.7/21.3 | SeqDiffuSeq | 61.4 | 17.2 | — | 92.7/— |
| AR-Diffusion | — | 31.7/10.1/24.7 | SSD-LM | 62.8 | 14.1 | **38.5** | 94.5/56.9 |
| GENIE | — | 29.3/8.3/21.9 | TESS (random) | 60.8 | 19.0 | 36.1 | **96.1**/62.4 |
| Embedding⋆ | 68.2 | 32.1/10.1/24.6 | Embedding⋆ | 62.0 | 18.9 | 35.2 | 92.4/61.2 |
| Simplex⋆ | 63.8 | 29.6/8.5/23.0 | Simplex⋆ | 63.0 | 19.3 | 36.9 | 93.0/63.8 |
| Smoothie⋆ (ours) | **68.8** | **33.7/11.1/26.0** | Smoothie⋆ (ours) | **63.1** | **19.9** | 36.5 | 92.8/63.3 |

**Baselines** We compare Smoothie against several diffusion-based and autoregressive baselines, all with approximately 100M parameters and trained from scratch on each dataset. The diffusion-based baselines include DiffuSeq (Gong et al., 2023a), SeqDiffuSeq (Yuan et al., 2022), SSD-LM (Han et al., 2023), TESS (Karimi Mahabadi et al., 2024), AR-Diffusion (Wu et al., 2023), and GENIE (Lin et al., 2023). We also compare against MDLM (Sahoo et al., 2024), an established masked diffusion model that we trained for sequence-to-sequence tasks using the provided code. For autoregressive baselines, we evaluate BART (Lewis et al., 2020), GPT-2 (Radford et al., 2019), GPVAE-T5 (Du et al., 2022), FLAN-T5 (Chung et al., 2024), and a standard Transformer (Vaswani et al., 2017). TESS approach uses pre-trained RoBERTa (Liu et al., 2019) to initialize its diffusion model. For a fair comparison, we only compare to the model trained from random initialization.

Additionally, we conduct a rigorous comparison of our proposed distance-based latent space with two previously explored alternatives: the embedding space (Gong et al., 2023a; Yuan et al., 2022) (Embedding⋆ in experiments) and the simplex space (Han et al., 2023; Karimi Mahabadi et al., 2024) (Simplex⋆ in experiments). To ensure a fair evaluation, we train all diffusion models under identical conditions, keeping the architecture, training hyperparameters, and decoding strategy fixed. The only variables are the latent space and its associated noise schedule. For embedding-based diffusion, we use the noise scheduler from Shabalin et al. (2025), while for simplex-based diffusion, we adopt the scheduler from Han et al. (2023). In all three cases, sampling is performed using a procedure defined in the respective latent space, following the formulation in Eq. 12. Smoothie and the embedding-based diffusion model are trained using MSE loss, while the simplex-based diffusion is trained using cross-entropy loss because it is not suitable for predicting continuous embeddings.

## 5.2 RESULTS

We now present a quantitative comparison of Smoothie against a range of generative models. Wherever possible, we adopt reported results from prior work (Karimi Mahabadi et al., 2024; Lovelace et al., 2023; Wu et al., 2023; Meshchaninov et al., 2025). When certain metrics are unavailable, we reproduce the corresponding methods using the original implementations. For consistency, we re-implement and train the embedding- and simplex-based diffusion baselines within our framework.

We show the results on XSum and Quasar-T dataset in Table 2, and on QQP, ParaDetox, and ROCStories in Table 3. Overall, Smoothie consistently outperforms other text diffusion approaches, as well as diffusion methods based on embedding- and simplex-based latent spaces, across all tasks, achieving quality comparable to that of autoregressive models.

Notably, embedding-based diffusion yields higher quality than simplex-based diffusion on all datasets except Quasar-T. This difference can be attributed to the fact that simplex-based diffusion does not incorporate semantic information into the noising process, making it inherently more chaotic. Nevertheless, when combined with our proposed architecture, simplex-based diffusion surpasses the TESS approach, which employs the same diffusion process and a training pipeline, differing only in

Table 3: Text generation results on ROCStories, QQP and ParaDetox datasets. † denotes autoregressive models, △ denotes the results reproduced with original code, ⋆ denotes our implementations. The best-performing *diffusion* results are highlighted in **bold**, the second-best are underlined.

| Method | ROCStories | | | QQP | | | | ParaDetox | |
|---|---|---|---|---|---|---|---|---|---|
| | MAUVE ↑ | PPL ↓ | Div ↑ | BS ↑ | BLEU ↑ | R-L ↑ | D-1/4 ↑ | BLEU ↑ | J-Score ↑ |
| GPT-2† | 78.9 | 20.5 | 25.2 | 82.5 | 19.8 | 52.1 | 98.0/62.5 | 67.7 | 60.4 |
| GPVAE-T5† | — | — | — | 84.7 | 24.1 | 58.9 | 96.9/61.7 | — | — |
| BART† | — | — | — | 85.7 | 30.4 | 61.4 | 98.8/61.0 | — | — |
| MDLM△ | 63.9 | 58.1 | **35.1** | 76.3 | 21.5 | 46.2 | 96.2/64.4 | 61.5 | 41.4 |
| DiffuSeq | 8.6 | 50.5 | 12.4 | 79.5 | 18.5 | — | 97.6/— | 67.9 | 47.5 |
| SeqDiffuSeq | 10.3 | 29.3 | 13.7 | 82.9 | 23.3 | — | 98.1/— | 68.8 | 48.6 |
| AR-Diffsion△ | 6.6 | 41.8 | 10.1 | 80.1 | 19.2 | 54.9 | — | 64.7 | 46.5 |
| SSD-LM | — | — | — | 83.8 | 22.9 | 58.3 | **98.8**/57.3 | – | – |
| Embedding⋆ | 23.4 | **18.6** | 13.6 | 83.4 | **31.3** | 59.4 | 97.7/64.5 | 67.6 | 49.1 |
| Simplex⋆ | 15.2 | 25.3 | 12.4 | 80.6 | 26.8 | 54.9 | 96.8/**64.8** | 65.1 | 47.7 |
| SMOOTHIE⋆ (ours) | **73.5** | 24.2 | 25.9 | **83.9** | 30.8 | **60.9** | 98.4/60.5 | **69.2** | **51.7** |

the architecture design. This highlights that selecting an appropriate model architecture is as critical as choosing the diffusion space.

The most pronounced improvement in generation quality is observed on the ROCStories dataset. By tuning the $\tilde{\delta}$ parameter (Section 5.1), SMOOTHIE effectively balances diversity and coherence, achieving the highest MAUVE score and nearly matching the quality of GPT-2. Note that embedding-based diffusion exhibits lower perplexity, primarily due to reduced diversity—a well-known limitation of the generative perplexity metric Holtzman et al. (2020).

### 5.3 AMOUNT OF DENOISING STEPS

Table 4 presents the relationship between the number of denoising steps and the generation quality of SMOOTHIE in terms of J-Score for ParaDetox and BERTScore for other datasets. We observe that for all datasets except ParaDetox, the quality does not change much regardless of the number of steps. Nevertheless, for XSum the performance improves as the number of steps increases until we reach 200 steps, after which the reprformance drops. This can be explained

Table 4: The impact of changing the number of steps on generation quality. We show J-score for ParaDetox and BERTScore for the other datasets.

| Steps | XSum | Quasar-T | QQP | ParaDetox |
|---|---|---|---|---|
| 25 | 67.7 | **63.1** | **83.9** | 51.1 |
| 50 | 68.5 | **63.1** | 83.8 | 51.4 |
| 100 | 68.7 | **63.1** | 83.7 | **51.7** |
| 200 | **68.8** | **63.1** | 83.6 | 51.0 |
| 500 | 68.4 | **63.1** | 83.5 | 50.8 |

by the impact of self-conditioning, which lead to a mismatch between train and generation trajectory for larger amount of steps Shabalin et al. (2025). Overall, the results align with the observation made in the TESS paper (Karimi Mahabadi et al., 2024), which suggests that the optimal number of denoising steps correlates with the complexity of the task.

## 6 CONCLUSION

In this work, we introduce SMOOTHIE, a text diffusion method that constructs its diffusion process with consideration of the discrete nature of text and the semantic relationships between tokens. To capture these properties, each token is mapped to a vector of Euclidean distances between its embedding and the embeddings of all tokens in the vocabulary. Our choice of the Euclidean distance is based on the Euclidean semantic space hypothesis (Hashimoto et al., 2016), which posits that semantic similarity correlates with Euclidean proximity in embedding space.

Our method also can be applicable to other categorical domains where semantic relationships exist between categories (e.g. graphs, protein sequences). However, in such cases, a different distance metric more suited to the domain's properties may be required. We leave the exploration of this direction to future work.

Empirical results on four sequence-to-sequence tasks demonstrate that SMOOTHIE outperforms existing text diffusion methods, as well as our diffusion model framework with alternative diffusion latent spaces that do not rely on additional encoders.

## REPRODUCIBILITY STATEMENT

To facilitate reproducibility, we release the source code used to train SMOOTHIE, as well as the embedding- and simplex-based diffusion models. A complete set of hyperparameter configurations is provided in Appendix E. All experiments are conducted exclusively on publicly available datasets, whose details are described in Appendix L.

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

## A  LIMITATIONS

**Pre-trained Embeddings**  Our proposed method relies on a pre-trained embedding matrix $\mathbf{E}$ from the BERT model. While this choice simplifies the training process and improves its stability, it limits the model's scalability and may cap its generation quality, because finetuning embeddings for a specific task should offer better results. An end-to-end training approach, as used in Li et al. (2022); Gong et al. (2023a); Karimi Mahabadi et al. (2024), could be applied to our method as well. We leave the exploration of this approach for future work.

**Fixed Sequence Length**  As with most text diffusion models, our method operates with a fixed sequence length. Variable-length outputs are emulated by discarding tokens past the end-of-sequence (EOS) token. This strategy introduces inefficiencies during training and generation, as the model must predict padding tokens regardless of actual sequence length. To the best of our knowledge, dynamically varying sequence lengths during the denoising stage remains an underexplored area. SeqDiffuSeq (Yuan et al., 2022) addresses this by truncating sequences early, based on the observation that the EOS token position often stabilizes early in denoising. However, this is an ad hoc solution, and more advanced approaches need to be developed.

## B  SOCIETAL IMPACT

Language models have been shown to produce harmful outputs (Weidinger et al., 2022), spread disinformation (Shao et al., 2018), hallucinate (Huang et al., 2025), and potentially violate user privacy (Carlini et al., 2021). Although our study focuses on tasks that differ from those typically used in prior harmfulness evaluations, future scaling of our approach could lead to similar negative outcomes. Research on methods for mitigating model harmfulness is actively developing, and we

believe that insights from this work may also inform improvements in the reliability and safety of text diffusion models.

## C    RELATIONSHIP BETWEEN DISTANCE-BASED AND SIMPLEX-BASED LATENT SPACES

In this section, we demonstrate that our proposed *distance-based latent space* generalizes the *simplex-based latent space*. Specifically, we show that the simplex-based latent space corresponds to a special case of a distance-based latent space when equipped with a trivial metric.

SMOOTHIE maps each token $w$ to a latent vector $\mathbf{d}^w$, where each component is given by:

$$\mathbf{d}^w_{(i)} = -\frac{1}{2}\|\mathbf{E}_w - \mathbf{E}_i\|^2. \tag{14}$$

For other categorical domains, the Euclidean distance can be replaced with a more suitable metric $\rho(w, i)$, leading to:

$$\mathbf{d}^w_{(i)} = -\rho(w, i). \tag{15}$$

To relate this to simplex-based representations, consider the case where $\rho$ is the *trivial metric*:

$$\rho(w, i) = [w \neq i], \tag{16}$$

i.e., $0$ when $w = i$ and $1$ otherwise. Under this choice, the latent vector becomes:

$$\mathbf{d}^w_{(i)} = \begin{cases} 0, & i = w, \\ -1, & \text{otherwise.} \end{cases} \tag{17}$$

In comparison, the simplex-based latent space maps each token $w$ to a vector $\mathbf{s}^w$ in the $k$-logit simplex:

$$\mathbf{s}^w_{(i)} = \begin{cases} +k, & i = w, \\ -k, & \text{otherwise.} \end{cases} \tag{18}$$

Both SMOOTHIE and simplex diffusion apply a Gaussian diffusion process to corrupt the latent vector:

$$\mathbf{z}_t = \phi_t \mathbf{z}_0 + \gamma_t \varepsilon, \tag{19}$$

where $\mathbf{z}_0 \in \{\mathbf{d}^w, \mathbf{s}^w\}$ and $\varepsilon \sim \mathcal{N}(0, I)$. To form a model input, the corrupted vector is then transformed into a probability distribution using the softmax function:

$$p_t = \text{softmax}(\mathbf{z}_t). \tag{20}$$

Since the softmax function is invariant to uniform additive shifts, we have:

$$\text{softmax}(\phi_t \mathbf{s}^w + \gamma_t \varepsilon) = \text{softmax}(\phi_t(\mathbf{s}^w - k) + \gamma_t \varepsilon) = \text{softmax}(2k\phi_t \mathbf{d}^w + \gamma_t \varepsilon), \tag{21}$$

where the final equality follows from observing that $\mathbf{s}^w - k = 2k\mathbf{d}^w$.

This confirms that the simplex-based latent space is equivalent, up to scaling, to the distance-based latent space under the trivial metric. Hence, the simplex-based representation is a special case within the more general distance-based latent space framework.

## D    PROOF OF THEOREM 4.1

*Proof.* We begin by recalling a standard result:

**Lemma.** The minimum value of the function $\mathbb{E}_{\mathbf{y}}\left[\|\mathbf{y} - \mathbf{z}\|^2\right]$ is achieved when $\mathbf{z} = \mathbb{E}[\mathbf{y}]$.

Using this lemma, we obtain:

$$g^*(\mathbf{p}_t, t) = \mathbb{E}_{\mathbf{w}^y}[\mathbf{D}_0(\mathbf{E}_{\mathbf{w}^y})] = \mathbb{E}_{\mathbf{w}^y}\left[ -\frac{1}{2}\big\{\|\mathbf{E}_{w_i^y} - \mathbf{E}_j\|^2\big\}_{i,j=1}^{m,V}\right] \quad \text{and} \quad f^*(\mathbf{p}_t, t) = \mathbb{E}_{\mathbf{w}^y}[\mathbf{E}_{\mathbf{w}^y}], \tag{22}$$

where $\mathbf{w}^y \sim p(\mathbf{w}^y \mid \mathbf{p}_t)$. Since both $g^*(\mathbf{p}_t, t)$ and $f^*(\mathbf{p}_t, t)$ are matrices, without loss of generality we will prove this statement for an arbitrary row $i$ and column $j$. For brevity, we will define $u = \mathbf{E}_{w_i^y}$ and $v = \mathbf{E}_j$. Then, we need to show that

$$\mathbb{E}_u\left[ -\frac{1}{2}\|u - v\|^2\right] = -\frac{1}{2}\|\mathbb{E}[u] - v\|^2 + C \tag{23}$$

Expanding both sides:

$$\mathbb{E}_u\left[\|u - v\|^2\right] = \mathbb{E}[\|u\|^2] - 2v^\top\mathbb{E}[u] + \|v\|^2$$
$$\|\mathbb{E}[u] - v\|^2 = \|\mathbb{E}[u]\|^2 - 2v^\top\mathbb{E}[u] + \|v\|^2$$

Subtracting:

$$\mathbb{E}[\|u\|^2] - \|\mathbb{E}[u]\|^2 = \sum_{k=1}^d \mathrm{Var}(u_k) =: C$$

Thus,

$$\mathbb{E}_u\left[-\frac{1}{2}\|u - v\|^2\right] = -\frac{1}{2}\|\mathbb{E}[u] - v\|^2 + \underbrace{-\frac{1}{2}C}_{\text{constant}},$$

where $C$ is a constant independent of $\mathbb{E}[u]$.

Since this holds for all $(i, j)$, the matrix identity holds:

$$g^*(\mathbf{p}_t, t) = \mathbf{D}_0(f^*(\mathbf{p}_t, t)) + \mathbf{C}$$

$\square$

## E   Implementation details

The hyperparemeters for training and inference of the models across all datasets are presented in Table 5. We trained our models using two 80 GB NVIDIA A100 GPUs for 15 hours on average. For all the tasks, we save checkpoints every 25,000 steps. We select the best checkpoint by the quality on the development set. During generation we do not apply the clamping trick (Li et al., 2022), since it does not improve quality in our experiments. We do not use the classifier-free guidance (Ho & Salimans, 2021) for the same reason.

## F   An impact of $\tilde{\delta}$ on seq2seq tasks

In this section, we measure how the quality of sequence-to-sequence generation changes when the value of $\tilde{\delta}$ varies. For this experiment, we consider values in the range of $\tilde{\delta} \in \{0.1, 0.25, 0.5, 0.75, 1\}$ and set the number of generation steps to 100. Table 6 reports J-Score for ParaDetox and BERTScore for all other datasets. Although the difference in quality for different $\tilde{\delta}$ is not as significant as for the unconditional generation, it can be seen that lower values of $\tilde{\delta}$ produce better quality overall. Following these results, we set $\tilde{\delta} = 0.1$ for all datasets.

Table 6: The impact of $\tilde{\delta}$ value on generation quality. We show J-Score for ParaDetox and BERTScore for the other datasets.

| $\tilde{\delta}$ | XSum | Quasar-T | QQP | ParaDetox |
|---|---|---|---|---|
| 0.1 | **68.8** | **63.1** | **83.7** | **51.7** |
| 0.25 | 68.7 | **63.1** | **83.7** | 51.4 |
| 0.5 | 68.7 | **63.1** | **83.7** | 51.3 |
| 0.75 | 68.6 | **63.1** | 83.6 | 50.9 |
| 1 | 68.2 | **63.1** | 83.4 | 50.9 |

Table 5: Complete hyperparameter configurations for all datasets.

| Hyperparameter | ROCStories | XSum | Quasar-T | QQP | ParaDetox |
|---|---|---|---|---|---|
| Tokenizer | | | `bert-base-cased` | | |
| Transformer Layers | | | 12 | | |
| Transformer Dim | | | 768 | | |
| Self-Attention Heads | | | 12 | | |
| Optimizer | | | AdamW | | |
| Learning Rate | | | $2 \cdot 10^{-4}$ | | |
| $\beta_1, \beta_2$ | | | 0.9, 0.98 | | |
| Warmup steps | | | 5000 | | |
| LR scheduler | | | Constant | | |
| Weight decay | | | 0.01 | | |
| Gradient clipping | | | 1 | | |
| EMA decay | | | 0.9999 | | |
| Batch size | 256 | 256 | 512 | 256 | 256 |
| Training steps | 1M | 225k | 150k | 50k | 150k |
| Max input length | — | 512 | 100 | 50 | 40 |
| Max target length | 80 | 64 | 50 | 50 | 40 |
| Generation steps | 350 | 200 | 100 | 25 | 100 |
| $d$ | 5 | 5 | 7 | 5 | 7 |
| $\delta, \sigma_{\min}, \sigma_{\max}$ | | | 1, 1.5, 200 | | |
| $\tilde{\delta}$ | 1.125 | 0.1 | 0.1 | 0.1 | 0.1 |

Table 7: Impact of self-conditioning on the generation performance on XSum, Quasar-T and QQP datasets.

| | XSum | | Quasar-T | | | QQP | | |
|---|---|---|---|---|---|---|---|---|
| Method | BS ↑ | R-L ↑ | BS ↑ | BLEU ↑ | R-L ↑ | BS ↑ | BLEU ↑ | R-L ↑ |
| Embedding | 68.2 | 24.6 | 62.0 | 18.9 | 35.2 | 83.5 | 31.6 | 59.6 |
| w/o SC | 65.2 | 23.6 | 62.9 | 19.5 | 36.0 | 81.7 | 27.7 | 57.4 |
| Simplex | 63.8 | 23.0 | 63.0 | 19.3 | 36.9 | 81.2 | 27.3 | 55.0 |
| w/o SC | 61.2 | 21.5 | 62.5 | 19.4 | 36.4 | 80.0 | 25.9 | 54.1 |
| SMOOTHIE | 68.8 | 26.0 | 63.0 | 19.0 | 35.8 | 83.9 | 30.8 | 60.9 |
| w/o SC | 67.5 | 25.4 | 61.9 | 19.0 | 35.7 | 83.2 | 29.4 | 59.9 |

## G  SELF-CONDITIONING

Previous studies have shown that self-conditioning significantly improves the quality of text diffusion models (Yuan et al., 2022; Shabalin et al., 2025; Karimi Mahabadi et al., 2024; Dieleman et al., 2022). In this section, we compare the performance of SMOOTHIE, as well as of embedding- and simplex-based diffusion models, with and without self-conditioning. The results on the XSum, Quasar-T, and QQP datasets are reported in Table 7. Although performance gains vary across models and datasets, self-conditioning generally improves quality, which confirms the previous observations.

## H  EMBEDDINGS ABLATION

Throughout this work, we utilize BERT embedding matrix to represent text tokens without additional comments. We find it important to evaluate the robustness of SMOOTHIE to other choices of embeddings. Therefore, we demonstrate how model performance changes on the ROCStories dataset when embeddings are changed. We choose two alternatives with the same hidden size: GPT-2 (Radford et al., 2019) embeddings with the vocabulary size of 50k

Table 8: The generation quality of SMOOTHIE trained with different embedding types on ROCStories dataset.

| Embeddings | MAUVE ↑ | PPL ↓ | Div ↑ |
|---|---|---|---|
| BERT (default) | 73.5 | 24.2 | 25.9 |
| GPT-2 | 64.4 | 23.1 | 25.0 |
| GloVe | 36.8 | 36.4 | 24.6 |

and GloVe (Pennington et al., 2014) embeddings trained manually on Wikipedia dataset for BPE tokens with the vocabulary size of 10k. In the Table 8 we show the results of the ablation.

In terms of perplexity and diversity, GPT-2 embeddings perform similarly to BERT, with the exception of MAUVE. However, these results are still better than of other methods (see Table 3). Interestingly, we found out that the optimal value of $\tilde{\delta}$ for GPT2 embeddings is lower than for BERT embeddings (1.03 vs 1.125). Most probably, this is because diversity increases naturally with the increase of the vocabulary size and the need to increase it artificially disappears. GloVe embeddings are worse than the ones extracted from a language model. Therefore, a significant drop in quality is not surprising. We can conclude that embeddings is an important component of the framework and the quality of the model does depend on the quality of embeddings. However, the method allows freedom in the choice of embeddings, which should help in applicability.

## I NOISE SCHEDULER ABLATION

In this work, we use a special *arctan* noise scheduler for SMOOTHIE to make sure that the model entropy grows linearly with $t$ (Dieleman et al., 2022). In this section, we perform an ablation study for the proposed noise scheduler by evaluating different values of $d$. In Table 9, we show the numerical performance on the ROCStories dataset. For each $d$ we chose the best $\tilde{\delta}$ based on MAUVE. Smaller $d$ values correspond to more aggressive corruption. The results suggest that while the difference is marginal, SMOOTHIE is sensitive to the choice of the noise scheduler.

Table 9: An impact of the parameter $d$ in noise scheduler on the generation quality on the ROCStories dataset.

|  | MAUVE ↑ | PPL ↓ | Div ↑ |
|---|---|---|---|
| $d = 4$ | 66.2 | 24.4 | 24.5 |
| $d = 5$ | 73.5 | 24.2 | 25.9 |
| $d = 6$ | 66.5 | 26.7 | 27.7 |
| $d = 7$ | 64.9 | 24.6 | 26.7 |

Figure 3 illustrates how the reconstruction loss and the accuracy of the predicted tokens depend on the timestep t for our noise scheduler. In other words, we evaluate how closely the prediction $\hat{x}_0 = f_\theta(\mathbf{p}_t, \mathbf{0}, t)$ matches the original $\mathbf{x}_0$. Accuracy is calculated only for non-padding tokens.

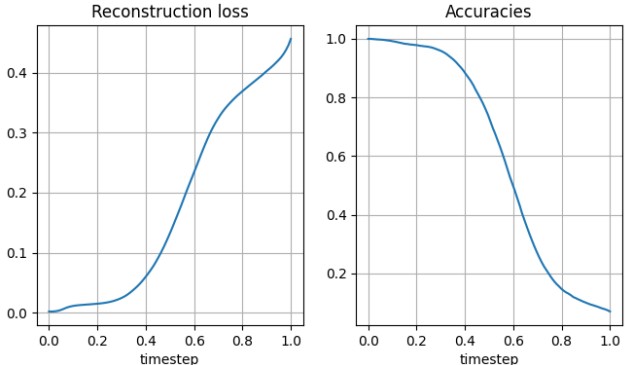

Figure 3: Reconstruction loss (left) and reconstruction accuracy (right) w.r.t. timestep for SMOOTHIE, trained with *arctan* noise scheduler with $d = 5$.

## J COMPUTATIONAL COMPLEXITY

In Table 10, we show the empirical comparison of runtime and GPU memory consumption for Smoothie, simplex- and embedding-based diffusions. We report the training time in hours for one million iterations on the ROCStories dataset. For generation, we perform 100 steps with a batch size of 32 and a sequence length of 80 and report the total generation time and the peak memory consumption.

We observe that our approach is slower than embedding-based diffusion, because we must compute pair-wise distances between sequence embeddings and vocabulary embedding on each iteration,

Table 10: The comparison of methods in terms of computational complexity.

| Method | Train time (h) | Generation time (s) | Memory (MB) |
|--------|---------------|---------------------|-------------|
| Embedding | 37.89 | 1.642 | 593.3 |
| Simplex | 51.91 | 2.897 | 678.4 |
| SMOOTHIE | 49.71 | 2.897 | 593.3 |

which have a complexity of $\mathcal{O}(\text{batch size} \times \text{seq len} \times d \times V)$. The generation is about $1.75\times$ slower, while training is $1.3\times$ slower. The difference in training speed is smaller because both methods involve gradient computation and parameter update with the same complexity. Simplex diffusion has approximately the same speed because it predicts tokens on each step instead of embeddings, which requires an application of a linear head with complexity $\mathcal{O}(\text{batch size} \times \text{seq len} \times d \times V)$ (same as SMOOTHIE). In terms of memory consumption all methods are the same, except simplex diffusion adds memory for storing the linear head. We also would like to note that several diffusion methods have the same exact complexity, because some of them utilize clamping trick (Li et al., 2022) and some predict tokens instead on embeddings, which requires an application of a large linear head (Han et al., 2023; Karimi Mahabadi et al., 2024; Dieleman et al., 2022).

## K  TRAINING DYNAMICS

In this section, we examine the differences in the training dynamics of SMOOTHIE, embedding and simplex diffusions. Figure 4 shows how training loss and Mauve change with respect to training time. For the embedding diffusion, we perform 1.75 times more generation steps (175 vs. 100) than for the other diffusion types, in order to match the generation time. The results suggest that, although SMOOTHIE trains and generates more slowly, it converges more quickly and produces higher-quality results than models trained for the same amount of time. We do not report loss for simplex diffusion, as it is trained with the cross-entropy loss, while other diffusions utilize MSE.

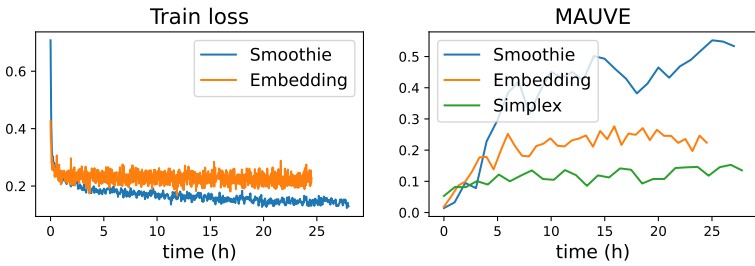

Figure 4: Training dynamics of SMOOTHIE, embedding and simplex diffusions on the ROCStories dataset.

## L  DATASET STATISTICS

**ROCStories**   The ROCStories dataset (Mostafazadeh et al., 2016) contains 98,161 five-sentence commonsense fictional stories that capture causal and temporal relations between everyday events. It is a widely used small-scale benchmark for unconditional text generation. The dataset is split into 93,161 training instances, 4,000 validation instances, and 1,000 test instances. Url: `https://cs.rochester.edu/nlp/rocstories/`

**XSum**   The XSum dataset (Narayan et al., 2018) is used for extreme summarization of BBC news articles. Each article covers a diverse range of topics (e.g., sports, politics) and is paired with a single-sentence summary. The dataset is divided into 204,045 training, 11,332 validation, and 11,334 test instances. Url: `https://huggingface.co/datasets/EdinburghNLP/xsum`

**Quasar-T**  Quasar-T (Dhingra et al., 2017) is a large-scale dataset for the question generation task. It requires models to comprehend natural language queries and extract answers from a large corpus. The dataset consists of open-domain trivia questions and their corresponding answers, collected from various internet sources. We use the version preprocessed by Gong et al. (2023a), which includes 116,953 training instances, 2,048 validation instances, and 10,000 test instances. Url: `https://github.com/Shark-NLP/DiffuSeq/tree/main`

**QQP**  The Quora Question Pairs (QQP) dataset (Chen et al., 2017) consists of over 400,000 question pairs from the Quora platform, each annotated with a binary label indicating whether the two questions are paraphrases. For the paraphrase generation task, we use the subset containing 149,263 positively labeled pairs, split into 119,410 training instances, 14,926 validation instances, and 14,927 test instances. Url: `https://huggingface.co/datasets/nyu-mll/glue/viewer/qqp`

**ParaDetox**  We use ParaDetox dataset (Logacheva et al., 2022) for small-scale conditional generation. It comprises 19,766 pairs of toxic and neutral comments and is intended for the text detoxification task. Url: `https://huggingface.co/datasets/s-nlp/paradetox`

