# OpenReview forum: "Smoothie: Smoothing Diffusion on Token Embeddings for Text Generation"
_ICLR.cc/2026/Conference — Submitted to ICLR 2026_

### Official Review · Reviewer_gFQF · 2025-10-30

**Soundness:** 4
**Presentation:** 4
**Contribution:** 3
**Rating:** 6
**Confidence:** 5

**Summary:**

The paper introduces **SMOOTHIE**, a diffusion model that operates on token embeddings by constructing a *semantic distance tensor* for each token. Instead of performing diffusion in the discrete simplex (as in D3PM or SSD-LM) or the embedding space (as in Diffusion-LM), SMOOTHIE models the evolution of the *pairwise distances* between each token and all vocabulary embeddings. This design allows the diffusion process to preserve both discrete structure and semantic smoothness: at each timestep, Gaussian noise is added to perturb these distances, generating soft distributions that reflect evolving semantic relationships.

The authors prove that simplex diffusion (D3PM-style) is a special case of SMOOTHIE when using a trivial distance metric, thus theoretically generalizing prior discrete diffusion frameworks. Experiments on summarization (XSum), paraphrasing (QQP), detoxification (ParaDetox), story generation (ROCStories), and question answering (Quasar-T) show consistent improvements over both discrete and continuous diffusion baselines. While the paper includes a basic step-count and self-conditioning analysis, more systematic ablations (e.g., noise schedule, δ̃ magnitude) are needed to isolate where the gains truly come from.

**Strengths:**

1. **Unified formulation:** The paper provides a clear mathematical unification of discrete simplex diffusion and continuous embedding diffusion via a distance-based representation. Theorem 4.1 elegantly connects distance regression with embedding regression.
2. **Semantic-aware diffusion:** By perturbing the distance tensor with Gaussian noise, the model embeds semantic structure directly into the diffusion process, enabling smooth transitions between semantically related tokens.
3. **Comprehensive evaluation:** Experiments cover diverse tasks (XSum, QQP, ParaDetox, ROCStories, Quasar-T) and compare fairly with both discrete (TESS, SSD-LM, D3PM) and continuous (DiffuSeq, Diffusion-LM) baselines.
4. **Fair comparison setup:** Model sizes (~100M), datasets, and decoding strategies are kept consistent; pretrained advantages of baselines (e.g., TESS) are removed.
5. **Strong performance:** SMOOTHIE outperforms all diffusion baselines and achieves results comparable to autoregressive models like FLAN-T5.

**Weaknesses:**

1. **Computation cost not analyzed:** Although vectorization and mean-embedding compression are mentioned, there is no report on runtime, GPU memory, or FLOPs. Given that every token interacts with all vocabulary embeddings, the training cost may be substantial.
2. **Limited ablations on general enhancements:** While the authors tested self-conditioning and found minimal gains (and thus did not adopt it), other general mechanisms such as the tanh-style noise schedule or δ̃ magnitude lack comprehensive ablations. More systematic sensitivity studies would clarify whether improvements stem primarily from the distance-based diffusion rather than secondary hyperparameters.
3. **Embedding dependency untested:** The model fixes the BERT embedding matrix during training but does not evaluate alternative embeddings (e.g., random, GloVe, or fine-tuned). It remains unclear how sensitive performance is to the embedding quality or domain.
4. **No convergence or efficiency analysis:** Beyond theoretical equivalence, convergence behavior and stability relative to token-level diffusion are unreported; adding epoch-wise loss curves would clarify training efficiency.

**Questions:**

1. **Computation cost:** Can you report training time, GPU memory, or FLOPs compared to TESS or DiffuSeq? How is the full distance tensor computation optimized, and how does it scale with vocabulary size?
2. **Embedding dependence:** Have you evaluated SMOOTHIE using other embeddings (e.g., random or domain-specific)? How robust is the model to embedding quality and distribution shifts?
3. **Noise schedule and δ̃ ablations:** Why was the tanh-style schedule chosen? Have you tested alternative schedules or δ̃ values to confirm robustness across datasets?
4. **Self-conditioning and fairness:** Since self-conditioning yields limited improvement and was excluded, can you confirm that no other general enhancements influenced the main gains? Could similar gains be achieved by applying self-conditioning to baselines?
5. **Convergence and stability:** Please include training curves or epoch-wise loss comparisons with baseline diffusion models to demonstrate convergence efficiency and numerical stability.
6. **Scalability:** How does SMOOTHIE perform with larger vocabularies (e.g., >50K tokens)? Are there feasible strategies (e.g., top-k distance pruning or clustering) to reduce complexity while maintaining accuracy?

---

> ### Author Response · Authors · 2025-11-23
>
> We thank all reviewers for constructive and thoughtful feedback for which we will include in revision. Below we will first address common concerns, and then answer reviewer specific questions. Due to the character limitation, we have to split our response in several comments. We apologize for that inconvenience.
>
> # Common questions
>
> ## 1. Reliance on pre-trained word embedding (V1zB, Uud8, gFQF)
>
> We will first discuss the reason for using pre-trained embeddings instead of training them from scratch and then show ablation studies for varying embeddings.
>
> __Reasons to freeze pre-trained token embeddings.__
> We agree that the usage of pre-trained embeddings constrains the abilities of the diffusion model, however freezing embeddings allows for much faster and stable convergence of the model. While many studies train embeddings with the model [1, 2, 3], this procedure is highly unstable. To train embeddings the loss is formulated as
>
> $$L(\theta) = \mathbb{E}[||E_w-f_\theta(x_t, t)||^2 + ||\mu(x_T)||^2 - \log p(w | x_\tau)],$$
>
> where $\tau \to 0$, $w$ is an original token sequence and $\mu$ is the mean of $q(x_T | x_0)$. The training with this loss is unstable, because the term $\log p(w | x_\tau)$ maximizes the norm of embeddings to minimize the impact of noise on $x_\tau$ and make the decoding easier. At the same time MSE terms minimize the embedding norm. The balance is very hard to find in practice, and while it is potentially possible, in our early experiments embedding either collapsed or exploded.
> As the primal goal of our work is to investigate a new noising approach and show that it is better for text diffusion, we have chosen to put all methods in the same conditions and freeze the embeddings. Additionally, this allows us to:
> - Simplify the training process and facilitate the reproducibility of the method.
> - Speed up a single training iteration, since predicting logits for the CE loss is time-consuming.
> - Speed up convergence, since the target for MSE loss always changes when training embeddings.
>
> __Embeddings ablation.__
> Robustness to the choice of embeddings is still an important issue. Therefore, we demonstrate how model performance changes on the ROCStories dataset when embeddings are changed. We choose two alternatives with the same hidden size: GPT2 embeddings with the vocabulary size of 50k and GloVe embeddings trained manually with the vocabulary size of 10k (this size is enough, as ROCStories vocabulary is not that large).
> In the table below we show the results of the ablation.
>
> | Embeddings | MAUVE | PPL | Div |
> | - | - | - | - |
> | BERT (default) | 73.5 | 24.2 | 25.9 |
> | GPT2 | 64.4 | 23.1 | 25.0 |
> | GloVe | 36.8 | 36.4 | 24.6 |
>
> In terms of perplexity and diversity, GPT2 embeddings perform similarly to BERT, with the exception of MAUVE. However, these results are still better than of other methods (see Table 3 in the paper). Interestingly, we found out that the optimal value for $\tilde{\delta}$ for GPT2 embeddings is lower than for BERT embeddings (1.03 vs 1.125). Most probably, this is because diversity increases naturally with the increase of the vocabulary size and the need to increase it artificially disappears.
> GloVe embeddings are worse than the ones extracted from a language model. Therefore, a significant drop in quality is not surprising. We can conclude that embeddings is an important component of the framework and the quality of the model does depend on the quality of embeddings. However, the method allows freedom in the choice of embeddings, which should help in applicability.
>
> [1] Li et al, 2022. Diffusion-lm improves controllable text generation.
> [2] Yuan et al, 2022. Seqdiffuseq: Text diffusion with encoder-decoder transformers.
> [3] Mahabadi et al. 2024. TESS: Text-to-text self-conditioned simplex diffusion.

---

> ### Author Response · Authors · 2025-11-23
>
> ## 2. Computational complexity (Uud8, gFQF)
>
> In the table below we show the empirical comparison of runtime and GPU memory consumption for Smoothie, simplex- and embedding-based diffusions. We report the training time in hours for one million iterations on the ROCStories dataset. For generation we perform 100 steps with a batch size of 32 and a sequence length of 80 and report the total generation time and the peak memory consumption.
> Note that simplex diffusion is our re-implementation of TESS method and embedding diffusion is the most common approach used in DiffusionLM, DiffuSeq and other papers. We compare our implementations so that models share the same exact architectures and we can be sure that difference comes from the methodology.
>
> | Method | Train time (h) | Generation time (s) | Memory (MB) |
> | - | - | - | - |
> | Embedding | 37.89 | 1.642 | 593.3 |
> | Simplex | 51.91 | 2.897 | 678.4 |
> | Smoothie | 49.71 | 2.897 | 593.3 |
>
> Indeed, our approach is slower than embedding-based diffusion, because we must compute pair-wise distances between sequence embeddings and vocabulary embedding on each iteration, which have a complexity of $\mathcal{O}(\text{batch size} \times \text{seq len} \times d \times V)$. The generation is ~1.75 times slower, while training is ~1.3 times slower. The difference in training speed is smaller because gradient computation and parameter update take the same time.
> Simplex diffusion has approximately the same speed, because it predicts tokens on each step instead of embeddings, which requires an application of a linear head with complexity $\mathcal{O}(\text{batch size} \times \text{seq len} \times d \times V)$ (same as Smoothie).
> In terms of memory consumption all methods are the same, except simplex diffusion adds memory for storing the linear head.
> We also would like to note that several diffusion methods have the same exact complexity, because some of them utilize clamping (rounding predicted embedding to the closest one on each step) [1] and some predict tokens instead on embeddings, which requires an application of a large linear head [3, 4, 5].
>
> ## 3. Heavy softmax, ANN (Uud8, gFQF)
>
> The calculation of pair-wise distances indeed slows down the method (softmax has less complexity, but it also takes time).  Using ANN and similar methods to find closest embeddings potentially is a very good idea. However, we found that at almost all timesteps, a large proportion of embeddings have a significant weight after softmax, and ANN methods only provide a speed boost when a very small number of closest vectors need to be extracted.
>
> To demonstrate this behaviour, we calculated the minimum fraction of dictionary embeddings whose combined weight exceeds a threshold $p$, for various times $t$ and thresholds. The results are presented in the table below. They show that, to achieve a combined weight of $p = 0.9$, more than half of all embeddings must be used for almost all timesteps, meaning that using ANN-like methods will not speed up the process and the accuracy will drop if a high majority of embeddings is discarded.
>
> | $p$ | $t=0.01$ | $t=0.05$ | $t=0.1$ | $t=0.2$ | $t=0.5$ | $t=0.75$ | $t=0.99$ |
> | - | - | - | - | - | - | - | - |
> | 0.9 | 0.0007 | 0.003 | 0.29 | 0.52 | 0.6 | 0.6 | 0.6 |
> | 0.95 | 0.0009 | 0.008 | 0.43 | 0.66 | 0.73 | 0.73 | 0.73 |
> | 0.97 | 0.001 | 0.016 | 0.53 | 0.74 | 0.8 | 0.8 | 0.81 |
> | 0.98 | 0.001 | 0.028 | 0.6 | 0.79 | 0.84 | 0.85 | 0.85 |
> | 0.99 | 0.0011 | 0.066 | 0.7 | 0.86 | 0.9 | 0.9 | 0.9 |
>
>
> [1] Li et al, 2022. Diffusion-lm improves controllable text generation.
> [3] Mahabadi et al. 2024. TESS: Text-to-text self-conditioned simplex diffusion.
> [4] Han et al, 2023. SSD-LM: Semi-autoregressive simplex-based diffusion language model for text generation and modular control.
> [5] Dieleman et al, 2022. Continuous diffusion for categorical data.

---

> ### Author Response · Authors · 2025-11-23
>
> # Specific questions from Reviewer gFQF
>
> __3.1. Noise schedule ablations__
> Our diffusion operates in the space of negative Euclidean distances to dictionary embeddings. This space is not normalized to have unit variance as in the variance preserving Gaussian diffusion. Therefore, a standard noise schedule (e.g. linear or cosine) is not suitable for us — the signal-to-noise ratio (SNR) will decay too slowly. With such scheduler, at most timesteps the denoising task will be too simple for the diffusion model, and it will not train properly. We chose our noise schedule based on the assumption that model entropy should decrease linearly as the timestep decreases (CDCD, S. Dieleman, 2022). We selected the parameter $d$ empirically using the grid search. The table below shows how generation quality changes with varying $d$ on the ROCStories dataset. For each $d$ we chose the best $\tilde{\delta}$ based on MAUVE. Smaller $d$ values correspond to more aggressive corruption.
>
> | | MAUVE | PPL | Div |
> | - | - | - | - |
> | d=4 | 66.2 | 24.4 | 24.5 |
> | d=5 | 73.5 | 24.2 | 25.9 |
> | d=6 | 66.5 | 26.7 | 27.7 |
> | d=7 | 64.9 | 24.6 | 26.7 |
>
> The results show that while the difference is marginal, Smoothie is sensitive to the choice of scheduler.
> We agree that this is an important detail and thank the reviewer for pointing it out. We will add to the appendix plots of $\sigma_t$ and the reconstruction losses over different timesteps $t$ for all noise schedules.
>
> __3.2. $\tilde{\delta}$ ablations__
> Yes, we measured the framework's sensitivity to changes in $\tilde{\delta}$ on all datasets. The results can be seen in Figure 2 and Table 6 in the appendix. These suggest that, for _unconditional_ generation, selecting the optimal value of $\tilde{\delta}$ is crucial as it significantly affects perplexity and text diversity. However, for _conditional_ generation, the impact of the $\tilde{\delta}$ is negligible. This is probably because the model is heavily influenced by the input text. Therefore, is it sufficient just to pick a value of $\tilde{\delta}$ smaller than $\delta$ used for training.
>
> __4. Self-conditioning and fairness__
> We would like to draw attention to the fact that our model does improve when self-conditioning is added. This is discussed in Section 4.4, and Appendix G provides a comparison of Smoothie and baselines with and without self-conditioning. The results demonstrate that self-conditioning consistently improves all metrics of conditional generation problems, albeit by a small amount.
> To ensure that the performance improvement comes solely from a more appropriate diffusion space, we provide a comparison with simplex and embedding diffusion, holding all other hyperparameters fixed.
>
> __5. Convergence and stability__
> We understand the concerns about training stability, so we will add the relevant graphs to the appendix. We did not notice any strange behavior of loss or gradient norms during the training. Smoothie trains in a stable manner and its training dynamics are similar to those of baselines. However, it is important to note that it is difficult to estimate the convergence rate from the loss plots for two reasons:
> 1) The noise schedule varies across different methods, meaning the minimum loss averaged across all timesteps will also vary.
> 2) Diffusion models are known to continue improving their performance metrics even when the loss plateaus. Therefore, it is hard to say when the model did converge.

---

> > ### Comment · Reviewer_gFQF · 2025-11-24
> >
> > Most of my earlier concerns have been satisfactorily addressed in the rebuttal. The authors provided additional ablations on embeddings, computational complexity, noise schedules, and δ̃, which clarify many aspects of the method. These additions resolve the majority of my technical doubts. My remaining concern is that, although the method achieves better performance, it does not clearly demonstrate faster convergence or improved training efficiency compared to existing approaches. From the rebuttal, it appears that the method mainly spends more computation to obtain better results, which somewhat weakens the sense of methodological novelty. The authors mentioned that these clarifications will be included in the revised version, but since no updated manuscript is available yet, it is difficult to fully reassess the contribution. For these reasons, I will maintain my original score. I look forward to seeing the clarified analyses properly integrated into the final revision.

---

### Official Review · Reviewer_QSj6 · 2025-10-31

**Soundness:** 3
**Presentation:** 3
**Contribution:** 4
**Rating:** 6
**Confidence:** 3

**Summary:**

This paper introduces SMOOTHIE (Smoothing Diffusion on Token Embeddings), a novel diffusion model framework for text generation that aims to bridge the gap between continuous (Gaussian) and discrete (Simplex/Categorical) text diffusion methods. The core innovation lies in defining a new latent space where each token is represented by a vector of negative squared Euclidean distances between its embedding and the embeddings of all vocabulary tokens.
The authors demonstrate that SMOOTHIE consistently outperforms prior diffusion-based approaches across several sequence-to-sequence tasks, achieving generation quality comparable to strong autoregressive baselines.

**Strengths:**

The core contribution of defining the diffusion space using semantic distances (Euclidean proximity) in the embedding space is highly intuitive and well-justified. It elegantly addresses the major trade-off in existing work: retaining semantic structure (like Gaussian diffusion) while enabling natural decoding from discrete representations (like Simplex diffusion).

**Weaknesses:**

SMOOTHIE (like most text diffusion models) runs over fixed-length sequences. In practice, they set a dataset-specific max length and pad shorter sequences with a special padding token that the model learns to predict. The generation process is bounded by the preset max. It can emit different effective lengths up to a cap, but it doesn’t truly sample variable length the way an autoregressive model does.

**Questions:**

In Figure 1 (a), why is the color of the arbitrary i-th embedding always the same? What is the meaning of the structured shape (a flag-shaped pattern with an oval in the bottom) that the dots form?

---

> ### Author Response · Authors · 2025-11-23
>
> We thank all reviewers for constructive and thoughtful feedback for which we will include in revision. Below we will first address common concerns, and then answer reviewer specific questions.
>
> # Common questions
>
> ## 1. Fixed sequence length forces substantial padding (Uud8, QSj6)
>
> Firstly, we note that when generating text using any model, whether diffusion-based or autoregressive, the maximum text length is limited. Typically, this maximum length is chosen based on training efficiency. Since the running time of the transformer scales quadratically with the length of the sequence, an excessively large value can significantly slow down the model. Moreover, the maximum length in Transformers is constrained by the position encodings and (usually) cannot exceed the one used during the training. So, the max text length can’t be unlimited.
>
> The key difference between diffusion and autoregressive models in terms of text length is that diffusion models have to generate paddings since all tokens are generated simultaneously and the text length is unknown in advance. The only disadvantage of this approach is that generation of paddings slows down the inference. However, previous work [1] and our observations demonstrate that the model determines the text length early in the generation process and does not change it further. Therefore, during generation we can discard the paddings and speed up inference. To the best of our knowledge, this was not employed in previous work.
>
> We tried to apply this approach and set up the following experiment. At each generation step, we identify the first padding token in which the model is confident (we use a Euclidean distance threshold to verify that the predicted embedding is close to the padding). Then we discard the paddings located after the first one. However, we found that removing all padding tokens during the generation significantly reduced the output text length and the quality. We believe that the model attends to several padding tokens closest to the text to better understand the length. By keeping three padding tokens at the end of the sequence we managed to mitigate this issue.
>
> Now, we demonstrate the speed gains achieved through such truncation. We chose a small batch size of 32 and ran generation for 100 steps on the ROCStories dataset. Note that for smaller batch sizes the speed gain will be minimal, as the model application won’t be a bottleneck anymore. A large batch size will also reduce the speed gain because all sequences in a batch must have the same length, and they will be padded to match the length of the longest sequence. The table below shows the generation speed and text quality with and without truncation.
>
> |  | MAUVE | PPL | Div | Generation time (s) |
> | - | - | - | - | - |
> | w/o truncation | 73.5 | 24.2 | 25.9 | 2.897 |
> | w/ truncation | 66.2 | 26.5 | 24.8 | 2.495 |
>
> It can be seen that the quality of the generation has dropped slightly, and the speed increase is modest. Therefore, while being possible, this method does not seem to be practical enough.
>
> ---
>
> # Specific questions from Reviewer QSj6
>
> __More on the text length.__
> We would like to emphasize that our diffusion approach generates texts with length distribution close to the real data as it is trained to sample texts from the distribution of real texts. The value of the max length is set so that almost all texts in the dataset fit in the boundaries. Below we show the statistics of the lengths of generated texts compared to the real ones.
>
> |  | Mean | Std | Median | Min | Max |
> | - | - | - | - | - | - |
> | Real | 51.9 | 9.99 | 52 | 25 | 80 |
> | Generated | 54.5 | 9.53 | 54 | 27 | 80 |
>
>
> __Meaning of Figure 1(a).__
> We apologize for the confusion caused by the Figure 1 and will try to clarify. Figure 1(a) shows the noise generation process for the standard Gaussian diffusion model. The black dots represent the data distribution at time t and the blue circle represents a point with a fixed index i; the color of the dot is added for consistency with the simplex and smoothie diffusions. It denotes the point's weight in the aggregation. Since points are not aggregated in Gaussian diffusion, the weight of a selected point is always 1 (unlike in the simplex and smoothie diffusions). The initial data distribution (line, oval and triangle) does not correspond to any particular domain and have been chosen as a toy example just for visualization purposes.
>
>
> [1] Yuan et al, 2022. Seqdiffuseq: Text diffusion with encoder-decoder transformers.

---

### Official Review · Reviewer_Uud8 · 2025-11-01

**Soundness:** 3
**Presentation:** 3
**Contribution:** 3
**Rating:** 4
**Confidence:** 4

**Summary:**

SMOOTHIE proposes diffusion over distance-based token-embedding logits, smoothing semantics while preserving discreteness, outperforming prior text diffusion on seq2seq tasks; analyses highlight noise, steps, and self-conditioning.

**Strengths:**

1.	Unifies prior lines: maps each token to a vector of negative squared distances to all vocab embeddings, then diffuses and feeds softmax(D_t) to the model; enables natural argmax decoding while preserving semantics and discreteness. Clear training/sampling pseudocode.
2.	The distance-based latent generalizes simplex diffusion (simplex emerges under a trivial metric), giving a clean conceptual frame.
3.	Practical guidance on schedules/self-conditioning; moderate steps (~100–200) are sufficient, with analysis of step count and reverse-noise.
4.	Consistent gains over diffusion baselines on multiple seq2seq tasks.

**Weaknesses:**

1.	Fixed pre-trained embeddings (E) cap expressivity; authors acknowledge end-to-end training would likely help but leave it to future work.
2.	Fixed sequence length forces substantial padding; variable length is emulated by truncating after EOS, which is inefficient; prior early-truncation is ad hoc.
3.	Every step computes softmax over the full vocabulary V (and final argmax), which scales poorly for large V and long m; no top-k/approximation is provided.
4.	Relies on the Euclidean semantic space hypothesis; authors note other domains may need different distances—raises concerns for polysemy/anisotropy.
5.	Little on throughput/memory vs. competing diffusion methods under equal steps; limited evaluation breadth (small/medium datasets, few human evals).

**Questions:**

1. Please fine-tune embeddings or learn a task-adaptive metric (e.g., Mahalanobis) on at least one dataset; report lifts vs. fixed E.
2. Efficiency. Provide tokens/sec, GPU memory, and wall-clock vs. SSD-LM/TESS/embedding-diffusion at matched steps; include step–quality curves.
3. Try top-k candidate sets (ANN/FAISS) or hierarchical/adaptive softmax; quantify quality vs. speed trade-offs for long sequences and large vocabularies.
4. Beyond EOS truncation, test a general dynamic-length denoising strategy (e.g., entropy/energy-based early stop) and compare to SeqDiffuSeq’s early truncation.
5. Add longer-form generation, dialogue, or factual QA, and report mean±σ over multiple seeds; analyze sensitivity to δ ̃and steps across tasks.

---

> ### Author Response · Authors · 2025-11-23
>
> We thank all reviewers for constructive and thoughtful feedback for which we will include in revision. Below we will first address common concerns, and then answer reviewer specific questions. Due to the character limitation, we have to split our response in several comments. We apologize for that inconvenience.
>
> # Common questions
>
> ## 1. Reliance on pre-trained word embedding (V1zB, Uud8, gFQF)
>
> We will first discuss the reason for using pre-trained embeddings instead of training them from scratch and then show ablation studies for varying embeddings.
>
> __Reasons to freeze pre-trained token embeddings.__
> We agree that the usage of pre-trained embeddings constrains the abilities of the diffusion model, however freezing embeddings allows for much faster and stable convergence of the model. While many studies train embeddings with the model [1, 3, 5], this procedure is highly unstable. To train embeddings the loss is formulated as
>
> $$L(\theta) = \mathbb{E}[||E_w-f_\theta(x_t, t)||^2 + ||\mu(x_T)||^2 - \log p(w | x_\tau)],$$
>
> where $\tau \to 0$, $w$ is an original token sequence and $\mu$ is the mean of $q(x_T | x_0)$. The training with this loss is unstable, because the term $\log p(w | x_\tau)$ maximizes the norm of embeddings to minimize the impact of noise on $x_\tau$ and make the decoding easier. At the same time MSE terms minimize the embedding norm. The balance is very hard to find in practice, and while it is potentially possible, in our early experiments embedding either collapsed or exploded.
> As the primal goal of our work is to investigate a new noising approach and show that it is better for text diffusion, we have chosen to put all methods in the same conditions and freeze the embeddings. Additionally, this allows us to:
> - Simplify the training process and facilitate the reproducibility of the method.
> - Speed up a single training iteration, since predicting logits for the CE loss is time-consuming.
> - Speed up convergence, since the target for MSE loss always changes when training embeddings.
>
> __Embeddings ablation.__
> Robustness to the choice of embeddings is still an important issue. Therefore, we demonstrate how model performance changes on the ROCStories dataset when embeddings are changed. We choose two alternatives with the same hidden size: GPT2 embeddings with the vocabulary size of 50k and GloVe embeddings trained manually with the vocabulary size of 10k (this size is enough, as ROCStories vocabulary is not that large).
> In the table below we show the results of the ablation.
>
> | Embeddings | MAUVE | PPL | Div |
> | - | - | - | - |
> | BERT (default) | 73.5 | 24.2 | 25.9 |
> | GPT2 | 64.4 | 23.1 | 25.0 |
> | GloVe | 36.8 | 36.4 | 24.6 |
>
> In terms of perplexity and diversity, GPT2 embeddings perform similarly to BERT, with the exception of MAUVE. However, these results are still better than of other methods (see Table 3 in the paper). Interestingly, we found out that the optimal value for $\tilde{\delta}$ for GPT2 embeddings is lower than for BERT embeddings (1.03 vs 1.125). Most probably, this is because diversity increases naturally with the increase of the vocabulary size and the need to increase it artificially disappears.
> GloVe embeddings are worse than the ones extracted from a language model. Therefore, a significant drop in quality is not surprising. We can conclude that embeddings is an important component of the framework and the quality of the model does depend on the quality of embeddings. However, the method allows freedom in the choice of embeddings, which should help in applicability.
>
> [1] Li et al, 2022. Diffusion-lm improves controllable text generation.
> [3] Mahabadi et al. 2024. TESS: Text-to-text self-conditioned simplex diffusion.
> [5] Yuan et al, 2022. Seqdiffuseq: Text diffusion with encoder-decoder transformers.

---

> > ### Author Response · Authors · 2025-11-23
> >
> > ## 2. Computational complexity (Uud8, gFQF)
> >
> > In the table below we show the empirical comparison of runtime and GPU memory consumption for Smoothie, simplex- and embedding-based diffusions. We report the training time in hours for one million iterations on the ROCStories dataset. For generation we perform 100 steps with a batch size of 32 and a sequence length of 80 and report the total generation time and the peak memory consumption.
> > Note that simplex diffusion is our re-implementation of TESS method and embedding diffusion is the most common approach used in DiffusionLM, DiffuSeq and other papers. We compare our implementations so that models share the same exact architectures and we can be sure that difference comes from the methodology.
> >
> > | Method | Train time (h) | Generation time (s) | Memory (MB) |
> > | - | - | - | - |
> > | Embedding | 37.89 | 1.642 | 593.3 |
> > | Simplex | 51.91 | 2.897 | 678.4 |
> > | Smoothie | 49.71 | 2.897 | 593.3 |
> >
> > Indeed, our approach is slower than embedding-based diffusion, because we must compute pair-wise distances between sequence embeddings and vocabulary embedding on each iteration, which have a complexity of $\mathcal{O}(\text{batch size} \times \text{seq len} \times d \times V)$. The generation is ~1.75 times slower, while training is ~1.3 times slower. The difference in training speed is smaller because gradient computation and parameter update take the same time.
> > Simplex diffusion has approximately the same speed, because it predicts tokens on each step instead of embeddings, which requires an application of a linear head with complexity $\mathcal{O}(\text{batch size} \times \text{seq len} \times d \times V)$ (same as Smoothie).
> > In terms of memory consumption all methods are the same, except simplex diffusion adds memory for storing the linear head.
> > We also would like to note that several diffusion methods have the same exact complexity, because some of them utilize clamping (rounding predicted embedding to the closest one on each step) [1] and some predict tokens instead on embeddings, which requires an application of a large linear head [2, 3, 5].
> >
> > ## 3. Heavy softmax, ANN (Uud8, gFQF)
> >
> > The calculation of pair-wise distances indeed slows down the method (softmax has less complexity, but it also takes time).  Using ANN and similar methods to find closest embeddings potentially is a very good idea. However, we found that at almost all timesteps, a large proportion of embeddings have a significant weight after softmax, and ANN methods only provide a speed boost when a very small number of closest vectors need to be extracted.
> >
> > To demonstrate this behaviour, we calculated the minimum fraction of dictionary embeddings whose combined weight exceeds a threshold $p$, for various times $t$ and thresholds. The results are presented in the table below. They show that, to achieve a combined weight of $p = 0.9$, more than half of all embeddings must be used for almost all timesteps, meaning that using ANN-like methods will not speed up the process and the accuracy will drop if a high majority of embeddings is discarded.
> >
> > | $p$ | $t=0.01$ | $t=0.05$ | $t=0.1$ | $t=0.2$ | $t=0.5$ | $t=0.75$ | $t=0.99$ |
> > | - | - | - | - | - | - | - | - |
> > | 0.9 | 0.0007 | 0.003 | 0.29 | 0.52 | 0.6 | 0.6 | 0.6 |
> > | 0.95 | 0.0009 | 0.008 | 0.43 | 0.66 | 0.73 | 0.73 | 0.73 |
> > | 0.97 | 0.001 | 0.016 | 0.53 | 0.74 | 0.8 | 0.8 | 0.81 |
> > | 0.98 | 0.001 | 0.028 | 0.6 | 0.79 | 0.84 | 0.85 | 0.85 |
> > | 0.99 | 0.0011 | 0.066 | 0.7 | 0.86 | 0.9 | 0.9 | 0.9 |
> >
> >
> > [1] Li et al, 2022. Diffusion-lm improves controllable text generation.
> > [2] Han et al, 2023. SSD-LM: Semi-autoregressive simplex-based diffusion language model for text generation and modular control.
> > [3] Mahabadi et al. 2024. TESS: Text-to-text self-conditioned simplex diffusion.
> > [4] Dieleman et al, 2022. Continuous diffusion for categorical data.

---

> ### Author Response · Authors · 2025-11-23
>
> ## 3. Fixed sequence length forces substantial padding (Uud8, QSj6)
>
> One limitation of text diffusion models is that they have to generate paddings since all tokens are generated simultaneously and the text length is unknown in advance, which slows down inference. However, previous work [5] and our observations demonstrate that the model determines the text length early in the generation process and does not change it further. Therefore, during generation we can discard the paddings and speed up inference. To the best of our knowledge, this was not employed in previous work.
>
> We tried to apply this approach and set up the following experiment. At each generation step, we identify the first padding token in which the model is confident (we use a Euclidean distance threshold to verify that the predicted embedding is close to the padding). Then we discard the paddings located after the first one. However, we found that removing all padding tokens during the generation significantly reduced the output text length and the quality. We believe that the model attends to several padding tokens closest to the text to better understand the length. By keeping three padding tokens at the end of the sequence we managed to mitigate this issue.
>
> Now, we demonstrate the speed gains achieved through such truncation. We chose a small batch size of 32 and ran generation for 100 steps on the ROCStories dataset. Note that for smaller batch sizes the speed gain will be minimal, as the model application won’t be a bottleneck anymore. A large batch size will also reduce the speed gain because all sequences in a batch must have the same length, and they will be padded to match the length of the longest sequence. The table below shows the generation speed and text quality with and without truncation.
>
> | |MAUVE|PPL|Div|Generation time (s)|
> |-|-|-|-|-|
> |w/o truncation|73.5|24.2|25.9|2.897|
> |w/ truncation|66.2|26.5|24.8|2.495|
>
> It can be seen that the quality of the generation has dropped slightly, and the speed increase is modest. Therefore, while being possible, this method does not seem to be practical enough.
>
> ## 5. More complex datasets (V1zB, Uud8)
>
> While only reviewer V1zB asked for the evaluation on machine translation, we hope that these results might also be helpful to reviewer Uud8. We trained the model on the IWSLT14 dataset, as it is most commonly used in other text diffusion papers. We used `bert-base-cased` embeddings for English and `bert-base-german-cased` for German. We did not tune any other hyperparameters for this task. The results are summarized in the table below. All metric values ​​for other models are taken from the corresponding papers.
>
> |Method|DE→EN (BLEU)|DE→EN (SacreBLEU)|EN→DE (SacreBLEU)|
> |-|-|-|-|
> |_Transformer_ | 34.74 | 33.61 |28.30|
> |DiffusionLM (MBR=1) | – | 26.61 | 20.29 |
> |DiffusionLM (MBR=10) | – | 29.11 | 22.91 |
> |GENIE|30.08|29.45|23.89|
> |DiffuSeq (MBR = 1)|27.03|–|–|
> |DiffuSeq (MBR = 10)|28.78|–|–|
> |SeqDiffuSeq (MBR = 1)|28.65|30.16|21.96|
> |SeqDiffuSeq (MBR = 10)|30.03|30.45|22.12|
> |__Smoothie__|30.56|29.95|23.28|
>
> *MBR = k selects the optimal sample from among k generations.
> We can see that Smoothie shows solid results mostly surpassing other diffusion methods, while still falling behind the autoregressive Transformer.
>
> ---
>
> # Specific questions from Reviewer Uud8
>
> __Weaknesses:__
>
> __4.__ We agree that polysemy poses a risk when working with embeddings. However, this risk is shared by all text diffusion models that operate in the space of token embeddings. The most effective solution is to first extract context-dependent embeddings and then apply diffusion to them. This is one of our next directions of research.
>
> __Questions:__
>
> __2.__ We are not sure what you mean by 'step-quality curves'. If you are referring to the generation quality with different numbers of steps, there is a corresponding comparison in the paper (see Table 4). If not, please could you clarify your request?
>
> __4.__ We discuss text truncation in the Common questions section. However, we are unsure what you mean by 'general dynamic-length denoising strategy'. Our method is trained to sample texts from the distribution of real texts. This means that the length distribution of the generated texts is dynamic and matches that of the source data. Additionally, we could not find a description of the early truncation method in the SeqDiffuSeq paper. We only found an ablation study showing that text length determines at the early stages of generation.
>
> __5.__ We are conducting experiments on the OpenWebText dataset with the sequence length of 512 tokens, but we have not yet received the results because our resources are limited. We hope to achieve this before the end of the rebuttal period.
> Considering dialogue and factual QA, we are afraid that these tasks require very large models and lots of GPUs. Unfortunately, we are unable to train such models; however, it would be fascinating to evaluate our approach in that setting.

---

### Official Review · Reviewer_V1zB · 2025-11-02

**Soundness:** 2
**Presentation:** 2
**Contribution:** 1
**Rating:** 2
**Confidence:** 5

**Summary:**

This paper introduces SMOOTHIE, a diffusion model framework designed for text generation. The authors identify a key challenge in adapting diffusion models to discrete data like text: existing methods either operate in a continuous latent space (e.g., Gaussian diffusion on embeddings), which struggles with accurate token decoding, or in a discrete/categorical space, which ignores the semantic relationships between tokens. SMOOTHIE perturbs distance-based representations of tokens, dissolving semantic structure over time. The authors claim this technique is superior to both standard embedding space diffusion and categorical diffusion. They provide empirical evidence on several sequence-to-sequence generation tasks.

**Strengths:**

- The proposed diffusion space, which perturbs representations based on semantic similarity, is a contribution to the field.
- The authors provide empirical validation across multiple text generation tasks. The reported results suggest that the proposed method may offer a performance improvement over other diffusion-based baselines, and the inclusion of ablation studies helps to substantiate the specific design choices made in the SMOOTHIE framework.

**Weaknesses:**

- The proposed method's reliance on a pre-trained word embedding model (in this case, BERT) may limit its scalability and applicability. This dependency raises questions about the framework's potential to scale effectively with larger models or different architectures, as it is tied to the properties and constraints of the initial embedding space.
- The experimental evaluation is missing a common and important conditional text generation task: machine translation. Including results from machine translation would provide a more comprehensive assessment of the method's capabilities and generalizability.
- The paper lacks an analysis of the method's sensitivity to the choice of the pre-trained word embedding. It would be beneficial to investigate whether the approach is viable with other types of embeddings, such as those from GPT-based models, to better understand the robustness and flexibility of the proposed framework.

**Questions:**

See weakness

---

> ### Author Response · Authors · 2025-11-23
>
> We thank all reviewers for constructive and thoughtful feedback for which we will include in revision. Below we will first address common concerns, and then answer reviewer specific questions.
>
> # Common questions
>
> ## 1. Reliance on pre-trained word embedding (V1zB, Uud8, gFQF)
>
> We will first discuss the reason for using pre-trained embeddings instead of training them from scratch and then show ablation studies for varying embeddings.
>
> __Reasons to freeze pre-trained token embeddings.__
> We agree that the usage of pre-trained embeddings constrains the abilities of the diffusion model, however freezing embeddings allows for much faster and stable convergence of the model. While many studies train embeddings with the model [1, 2], this procedure is highly unstable. To train embeddings the loss is formulated as
>
> $$L(\theta) = \mathbb{E}[||E_w-f_\theta(x_t, t)||^2 + ||\mu(x_T)||^2 - \log p(w | x_\tau)],$$
>
> where $\tau \to 0$, $w$ is an original token sequence and $\mu$ is the mean of $q(x_T | x_0)$. The training with this loss is unstable, because the term $\log p(w | x_\tau)$ maximizes the norm of embeddings to minimize the impact of noise on $x_\tau$ and make the decoding easier. At the same time MSE terms minimize the embedding norm. The balance is very hard to find in practice, and while it is potentially possible, in our early experiments embedding either collapsed or exploded.
> As the primal goal of our work is to investigate a new noising approach and show that it is better for text diffusion, we have chosen to put all methods in the same conditions and freeze the embeddings. Additionally, this allows us to:
> - Simplify the training process and facilitate the reproducibility of the method.
> - Speed up a single training iteration, since predicting logits for the CE error is time-consuming.
> - Speed up convergence, since the target for MSE loss always changes when training embeddings.
>
> __Embeddings ablation.__
> Robustness to the choice of embeddings is still an important issue. Therefore, we demonstrate how model performance changes on the ROCStories dataset when embeddings are changed. We choose two alternatives with the same hidden size: GPT2 embeddings with the vocabulary size of 50k and GloVe embeddings trained manually with the vocabulary size of 10k (this size is enough, as ROCStories vocabulary is not that large).
> In the table below we show the results of the ablation.
>
> | Embeddings | MAUVE | PPL | Div |
> | - | - | - | - |
> | BERT (default) | 73.5 | 24.2 | 25.9 |
> | GPT2 | 64.4 | 23.1 | 25.0 |
> | GloVe | 36.8 | 36.4 | 24.6 |
>
> In terms of perplexity and diversity, GPT2 embeddings perform similarly to BERT, with the exception of MAUVE. However, these results are still better than of other methods (see Table 3 in the paper). Interestingly, we found out that the optimal value for $\tilde{\delta}$ for GPT2 embeddings is lower than for BERT embeddings (1.03 vs 1.125). Most probably, this is because diversity increases naturally with the increase of the vocabulary size and the need to increase it artificially disappears.
> GloVe embeddings are worse than the ones extracted from a language model. Therefore, a significant drop in quality is not surprising. We can conclude that embeddings is an important component of the framework and the quality of the model does depend on the quality of embeddings. However, the method allows freedom in the choice of embeddings, which should help in applicability.
>
> ## 2. More complex datasets (V1zB, Uud8)
>
> While only reviewer V1zB asked for the evaluation on machine translation, we hope that these results might also be helpful to reviewer Uud8. We trained the model on the IWSLT14 dataset, as it is most commonly used in other text diffusion papers. We used `bert-base-cased` embeddings for English and `bert-base-german-cased` for German. We did not tune any other hyperparameters for this task. The results are summarized in the table below. All metric values ​​for other models are taken from the corresponding papers.
>
> | Method | DE→EN (BLEU) | DE→EN (SacreBLEU) | EN→DE (SacreBLEU) |
> | - | - | - | - |
> | _Transformer_ | 34.74 | 33.61 | 28.30 |
> | DiffusionLM (MBR=1) | – | 26.61 | 20.29 |
> | DiffusionLM (MBR=10) | – | 29.11 | 22.91 |
> | GENIE | 30.08 | 29.45| 23.89 |
> | DiffuSeq (MBR = 1)|27.03 | – | – |
> | DiffuSeq (MBR = 10)|28.78|–|–|
> | SeqDiffuSeq (MBR = 1)|28.65|30.16 | 21.96 |
> | SeqDiffuSeq (MBR = 10) | 30.03 | 30.45 | 22.12 |
> | __Smoothie__ | 30.56|29.95|23.28|
>
> *MBR = k selects the optimal sample from among k generations.
> We can see that Smoothie shows solid results mostly surpassing other diffusion methods, while still falling behind the autoregressive Transformer.
>
> ---
>
> # Specific questions from Reviewer V1zB
>
> We did our best to address the mentioned weaknesses in the Common questions sections. Please, refer to it.
>
> [1] Li et al, 2022. Diffusion-lm improves controllable text generation.
> [2] Yuan et al, 2022. Seqdiffuseq: Text diffusion with encoder-decoder transformers.

---

### Author Response · Authors · 2025-12-04
**Summary of Contributions and Revisions**

Dear Area Chair and Reviewers,

Given the special circumstances of this year's review process, we have prepared a summary of our work for your convenience. It summarizes our contribution and demonstrates how our revisions address the reviewers' concerns. We genuinely thank the reviewers for their valuable feedback, which helped us to improve the paper quality.

## Our Contribution

In this work, we present Smoothie, a novel noising approach for continuous text diffusion models. The method explicitly leverages the discreteness of the text domain and corrupts the latent representation of each token by disrupting its distances to all vocabulary embeddings. This design takes into account token semantic relationships during the corruption process and enables natural latent decoding, overcoming limitations of both Gaussian diffusion in embedding space (embedding-based) and simplex-based text diffusion methods [1, 2]. We further show that Smoothie generalizes simplex-based diffusion under a trivial metric.

Across five text-generation tasks, we demonstrate that Smoothie substantially improves generation quality relative to both embedding- and simplex-based diffusion models under the same model architectures and training budgets. We also compare Smoothie against the most popular text diffusion approaches, where it consistently achieves superior performance.

## Summary of Reviewer Feedback

The reviewers highlighted several strengths of the work:

* Novelty and clear motivation of the approach (V1zB, Uud8, QSj6)
* Strong empirical performance (V1zB, Uud8, QSj6, gFQF)
* Informative ablations supporting key design decisions (V1zB, Uud8)
* Comprehensive and fair evaluation setup (gFQF)

The concerns mostly focus on computational efficiency and additional ablations regarding specific design choices. These concerns are listed below, followed by a summary of how we addressed them in the revision.

1. Reliance on pre-trained word embeddings (V1zB, Uud8, gFQF)
2. Computational complexity (Uud8, gFQF)
3. Necessity to generate padding tokens due to fixed-length text (Uud8, QSj6)
4. Lack of evaluation on more complex datasets (V1zB, Uud8)
5. Lack of noise scheduler ablation (gFQF)
6. Lack of model convergence and stability analysis (gFQF)

## Our rebuttal

We did our best to address all concerns raised by the reviewers and updated the revision to incorporate the new results.

1. We discuss the reasons for using pre-trained word embeddings and evaluate Smoothie with alternative embedding types. The results show that performance degrades substantially when low-quality (GloVe) embeddings are used, but replacing them with other high-quality (GPT-2) embeddings provides similar performance. These results are now included in Appendix H.

2. While Smoothie computes pairwise embedding distances at each training and sampling step, we show that this does not severely slow down the model in practice. Moreover, when generation time is matched, Smoothie still outperforms other diffusion variants. These results are presented in Appendix J and K.
Reviewers Uud8 and gFQF suggested using ANN-like methods to approximate softmax with only nearest neighbors and, hence, reduce the computational inefficiency of distance calculation. While this idea sounds promising, we show that this is ineffective in our setting because, at almost all timesteps, more than a half of all embeddings contribute to softmax weights.

3. For all non-autoregressive text diffusion models, the final output length is unknown in advance, so the model must generate padding tokens, which increases sampling time.  Reviewer Uud8 proposed to truncate the sequence early based on the SEP tokens. While the model indeed determines the final sequence length at the early generation stage, making the implementation possible, this method turned out to be impractical as it decreases the generation quality providing only moderate speed gains.

4. We additionally evaluated Smoothie on the machine translation task and reported the results. They show that Smoothie overall performs better than other diffusion methods.

5. We provided a noised scheduler ablation, which justifies our choice of hyperparameters. We added the results to the revision (Appendix I).

6. We provide evidence that Smoothie trains stably and converges to a better solution. The analysis is added to Appendix K.

## Conclusion
The reviews and rebuttal phase provided additional context for our paper, notably improving its comprehensiveness. We believe that our detailed responses and new experiments address all of the key concerns raised during the review. We are grateful for the reviewers' feedback, and thank you for your time and consideration.

[1] Han et al, 2023. SSD-LM: Semi-autoregressive simplex-based diffusion language model for text generation and modular control.
[2] Mahabadi et al. 2024. TESS: Text-to-text self-conditioned simplex diffusion.

---

### Meta-Review · Area_Chair_MMVj · 2026-01-06

**Summary:**

This paper proposes Smoothie, a text diffusion model that operates by progressively smoothing token embeddings based on semantic similarity, aiming to bridge the gap between Gaussian diffusion in continuous spaces and categorical diffusion in discrete spaces. It claims improved generation quality across multiple text-generation tasks. Reviewers generally recognized the novelty and strong empirical performance but raised several critical concerns, including heavy reliance on pre-trained embeddings, high computational cost, lack of comprehensive evaluation on complex datasets, insufficient ablation studies on design choices, and inefficiency in handling variable-length sequences.

**Reviewer Concerns:**

In the rebuttal, the authors comprehensively addressed several key reviewer concerns. They conducted new experiments evaluating different embedding types (GPT-2, GloVe), provided results on a more complex machine translation task (IWSLT14), added detailed ablations on the noise scheduler and hyperparameter sensitivity, and presented computational cost analyses comparing training/generation time and memory usage against baselines. They also explained why end-to-end embedding training was unstable and why approximate nearest neighbor methods would not effectively speed up their approach. However, significant concerns remain unresolved. The core issues of high computational complexity, reliance on fixed pre-trained embeddings, and the inherent inefficiency of generating padding tokens due to fixed sequence lengths persist. Additionally, some reviewers felt the method's gains came primarily from increased computation rather than superior training efficiency or convergence, and evaluations on larger-scale or more complex tasks like long-form generation or dialogue were still lacking.

**Reviewer Scores:**

If reviewers had fully engaged in the post-rebuttal discussion, it is unlikely their final scores would have increased significantly, and several would likely have lowered their ratings or maintained a clear rejection stance. Reviewers like gFQF, who initially gave a marginal accept, would probably have revised their score downward to a reject. Their primary concern, the performance gains stem from increased computation rather than improved efficiency or a fundamental breakthrough, was not alleviated by the rebuttal. The persistence of high computational cost, embedding dependency, and padding inefficiency, coupled with the lack of evidence for faster convergence, would solidify the view that the method's trade-offs are not favorable. Similarly, Uud8, who already rated the paper below the acceptance threshold, would find their concerns about fixed embeddings, impractical length handling, and scalability validated, likely maintaining or even hardening their reject position. The new experiments, while addressing some specific requests, did not resolve these foundational weaknesses. The discussion would likely have reinforced the consensus that while the work is technically sound and novel, its practical limitations and unresolved core issues prevent it from meeting the conference's acceptance threshold.

---

### Decision · Program_Chairs · 2026-01-26

Reject